# Divergent urban storm response to convective, frontal and tropical systems

Xinxin Sui[1,2]✉, John Nielsen-Gammon[3], Zong-Liang Yang[4] & Dev Niyogi[1,4,5]

Urbanization modifies precipitation[1,2], yet previous studies have reported inconsistent results, with some cities experiencing rainfall enhancement and others showing suppression[3]. To reconcile these discrepancies, we examine how urban impacts vary across storm types using an event-based analysis. With three-dimensional radar reflectivity data (1995–2017), we identify more than 40,000 warm-season storms across four Texas cities (Dallas, Austin, San Antonio and Houston). Here we show that classifying storms into five types reveals distinct urban influences linked to storm scales and dynamics. Local-scale single-cell and isolated storms, driven by atmospheric instability, increase in frequency (7–31%), particularly at night. Synoptic-scale frontal storms show unchanged occurrence but contrasting intensity responses: cold fronts weaken over cities by 16–28%, probably because of thermal and roughness effects, whereas warm fronts exhibit enhanced reflectivity. Tropical systems show no consistent change in frequency or intensity but exhibit a shift of high-reflectivity grid cells towards lower altitudes over urban areas. Given the diverse climate and geography of Texas, this work provides a transferable framework for understanding urban–storm interactions in other regions. These findings move beyond the traditional 'urban wet or dry islands' model, advancing our understanding of how urbanization modulates extreme precipitation and informing climate modelling[4,5] and resilience planning for rapidly growing cities[6,7].

Urbanization profoundly alters regional climate and extreme weather. For instance, heatwaves, intensified by urban heat islands, increase health risks and mortality in cities[8]. Urban areas also affect regional rainfall patterns and trigger flash floods, causing substantial loss of life and property[6,7]. Although the temperature effects of urbanization are relatively well understood, its influence on rainfall is more complex and dynamic, which remains insufficiently resolved in present climate and meteorological models[4,5]. This gap leads to inaccurate predictions of urban precipitation, a critical concern given the dense populations and the vulnerability to extreme events in cities. Previous large-scale analyses show that cities and their downwind regions receive more annual rainfall than rural surroundings[2,3]. Many previous studies have also examined storm-specific urban impacts using numerical simulations, particularly for convective rainfall[9–11] and tropical cyclones[12,13]. However, variations across storm types and their underlying mechanisms remain underexplored in large observational samples.

Case studies, including both observations and model simulations across cities worldwide[14,15], have identified several mechanisms influencing precipitation, such as urban heat islands[1], surface roughness[16,17] and anthropogenic aerosols[18]. Statistical analyses of observations reveal a close relationship between the magnitude of urban heat islands and rainfall enhancement over cities[3]. The relationship arises because urban heat islands increase atmospheric instability[11], which enhances upward convection and promotes convective rainfall[9]. Under weak wind conditions, rainfall enhancement tends to occur over cities, whereas strong winds shift rainfall maxima to downwind areas[19].

Much of this observational understanding has been derived from accumulated rainfall comparisons based on long-time climatological records[20] rather than individual precipitation events[21,22]. However, because different weather systems are governed by distinct physical processes, urbanization may influence different storm types in different ways. Moreover, certain storm types contribute disproportionately to extreme precipitation. Therefore, it is essential to distinguish storm types and examine the specific effects of urbanization on each type. Tropical systems, spawned offshore, deliver large volumes of rainfall and cause serious disasters worldwide. Frontal storms, driven by thermal gradients, are also common and can produce intense precipitation. As well as these larger-scale systems, more frequent isolated convective storms can generate short-duration, high-intensity rainfall over localized areas. Previous studies have used numerical weather prediction models to simulate the impacts of urbanization on individual storms with different mechanisms[10,23]. However, owing to urban-resolving limitations and parameter sensitivity[24], these simulations often struggle to establish general characteristics of urban impacts, even with expensive ensemble approaches. Therefore, more observational evidence across a large number of storm cases is needed.

Texas, the largest state in the contiguous United States, contains four of the nation's ten most populous cities: Houston, Dallas, Austin

[1]Fariborz Maseeh Department of Civil, Architectural and Environmental Engineering, Cockrell School of Engineering, The University of Texas at Austin, Austin, TX, USA. [2]Civil and Environmental Engineering, Colorado School of Mines, Golden, CO, USA. [3]Department of Atmospheric Sciences, Texas A&M University, College Station, TX, USA. [4]Department of Earth and Planetary Sciences, Jackson School of Geosciences, The University of Texas at Austin, Austin, TX, USA. [5]Oden Institute for Computational Engineering and Sciences, The University of Texas at Austin, Austin, TX, USA. ✉e-mail: xinxin.sui@mines.edu

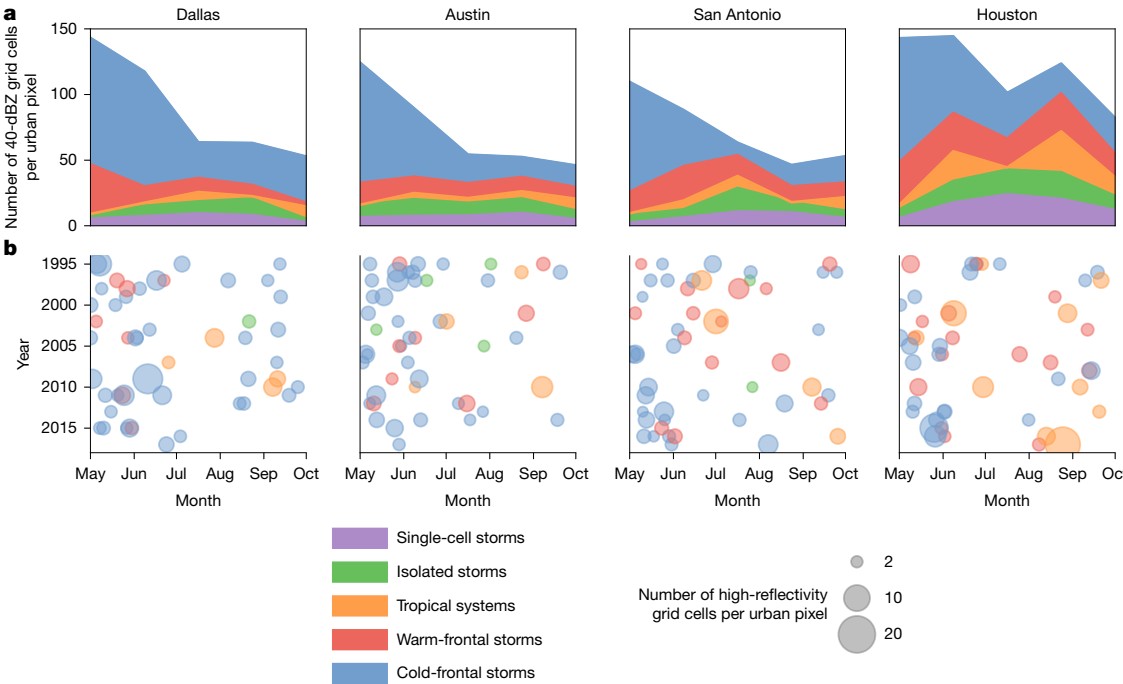

**Fig. 1 | Storm-type contributions shape temporal patterns of heavy rainfall in four Texas cities. a**, Occurrence of high-reflectivity grid cells (≥40 dBZ) contributed by five storm types, aggregated over 23 warm seasons across all altitudes (1995–2017). The storm types include single-cell storms, isolated storms, tropical systems, warm-frontal storms and cold-frontal storms. The corresponding contributions of ≥40 dBZ (and ≥20 dBZ) grid cells from each storm type are summarized in Extended Data Table 3. **b**, Monthly and interannual distributions of the 50 most intense storms per city, ranked by the number of high-reflectivity grid cells over urban areas during each storm event.

and San Antonio. Their rapidly growing population and complex meteorological setting from the dry line to the Gulf Coast make them particularly susceptible to extreme weather and natural hazards, making Texas cities ideal for investigating a wide range of storm types. In this study, we extract more than 40,000 storm events from a 23-year gridded radar dataset, GridRad[25], and examine the urbanization effects on individual storms across four cities. Because previous studies have identified stronger urban effects on summer precipitation than in other seasons[26], we focus on intense warm-season (May–September) storms. We classify the extracted storms into five types, local-scale single-cell and isolated storms, cold-frontal and warm-frontal storms, and tropically influenced weather systems, to investigate how urban influences vary across storm types. Given that the four Texas cities span diverse climatological and geographical settings, this work provides methodological insights and knowledge generalization applicable to other regions.

## Warm-season storms in Texas cities

To examine the three-dimensional structure of storms, we use GridRad radar reflectivity data in this research[25]. We focus on the storms that produce heavy rainfall over urban domains during the warm season (May–September), selecting events in which high-reflectivity areas (the column-maximum reflectivity exceeding 40 dBZ, approximately 10 mm h⁻¹ precipitation intensity) cover at least 20% of the urban surface. To capture the process of storms approaching and moving away from cities, the observation window extends twice the urban footprint beyond city boundaries (Extended Data Fig. 1). In this way, we identify approximately 1,000 storm events for San Antonio and Austin, about 2,000 storm events for Dallas–Fort Worth, and roughly 5,000 for Houston (Extended Data Table 1). These counts reflect both regional precipitation frequency and domain size, with more storms detected over larger cities. We classify all warm-season storms into five types

based on storm structure and motion (Extended Data Table 2), assigning names that reflect their likely forcing mechanisms. Most events are localized single-cell storms, ranging from 55.5% of all storms in Austin to 81.2% in Houston. Cold-frontal storms (4.7–18.7%) and regional isolated storms (9.8–16.8%) also constitute considerable fractions, whereas warm-frontal storms (3.7–9.8%) and tropical systems (0.5–1.7%) are relatively less frequent (Extended Data Table 1).

Despite their high frequency, single-cell storms, because of their short duration and limited spatial extent, contribute only 7.5% (Dallas) to 13.5% (Houston) of the total number of high-reflectivity grid cells (≥40 dBZ) across all altitudes (Extended Data Table 3). Cold-frontal storms produce the largest share of high-reflectivity grid cells (40.0–63.2%), particularly in northern regions. Warm-frontal storms contribute the second-largest proportion (16.4–22.7%), whereas isolated storms (8.2–13.5%) and tropical systems (4.6–12.3%) contribute less. We further calculate the average duration and high-reflectivity areas for each storm type passing through the observation window (Extended Data Table 4). Localized single-cell storms and isolated storms exhibit similar characteristics across all four cities. Single-cell storms last, on average, about 2 h and reach a maximum high-reflectivity areas of 50 km² during their lifetime, whereas isolated storms persist roughly three times longer and have heavy rain areas about eight times larger. Statistics for larger-scale systems, such as frontal and tropical storms, are more constrained by the size of the observation window, leading to longer durations and larger high-reflectivity areas over larger cities. Cold-frontal storms typically persist for 12–20 h and produce peak high-reflectivity areas of 3,000–6,400 km², whereas warm-frontal storms tend to last slightly longer (15–19 h) but generate smaller high-reflectivity areas (1,800–4,600 km²). Tropical systems are the most intense among the five storm types, with the most high-reflectivity grid cells; they can generate continuous rainfall over several days and high-reflectivity areas larger than 9,000 km² in Houston on average.

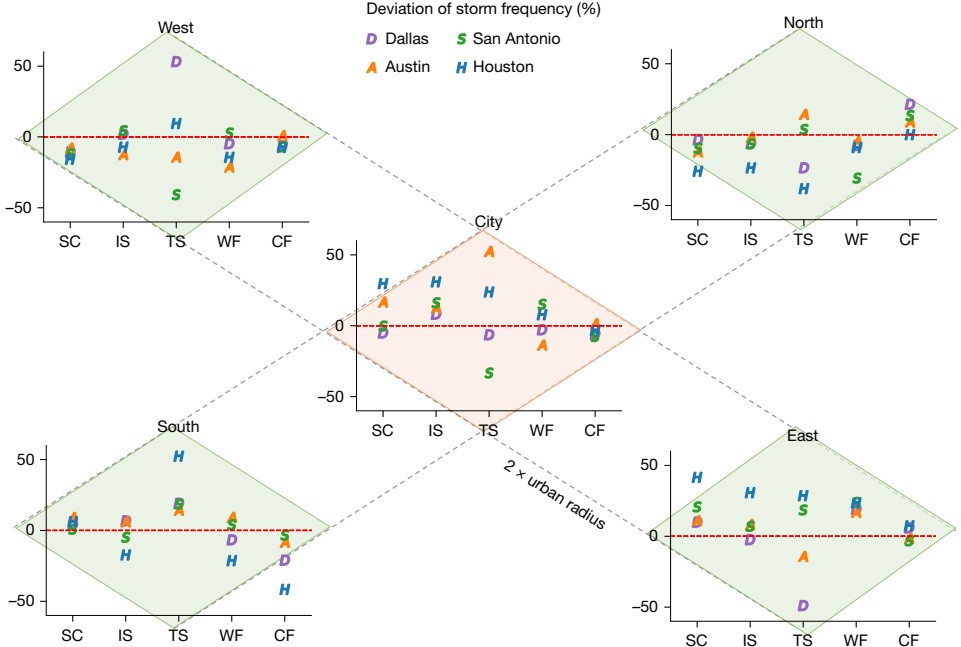

**Fig. 2 | Urban–rural differences in storm occurrence vary across storm types.** Deviations in the number of five types of warm-season storm in four Texas cities relative to the average number in their peripheral rural control areas (1995–2017). Paired Wilcoxon signed-rank tests on annual storm counts ($n = 92$) confirm that single-cell storms (SC) and isolated storms (IS) occur significantly more frequently in urban areas than in their rural counterparts, whereas no significant differences are detected for the other storm types (Supplementary Table 1). A similar analysis of deviations in the number of high-reflectivity grid cells ($\geq 40$ dBZ) contributed by each storm type is shown in Extended Data Fig. 2. CF, cold-frontal storms; TS, tropical systems; WF, warm-frontal storms.

We further investigate the monthly and interannual distributions of these storms over the 23-year study period (Fig. 1). Among the five warm-season months, May is the wettest, with the highest numbers of high-reflectivity grid cells across all levels and the highest frequency of extreme rainfall events, largely contributed by relatively frequent and intense cold-frontal storms. Local-scale single-cell storms and isolated storms are most common during the hotter months of July and August. In southern cities, warm-frontal storms and tropical systems are more prevalent and contribute more substantially to total rainfall, particularly in the coastal city of Houston along the Gulf Coast. The most extreme storm observed in Houston during the study period was Hurricane Harvey in August 2017, which is identified here as a tropical system (Fig. 1b).

## More local-scale storms over urban areas

To identify different storm performances in urban areas from surrounding rural environments, we define four rural comparison domains (north, south, west and east) around each city and extract storms passing through these rural domains using the same method. Spatial variations in storm counts (Fig. 2) and high-reflectivity grid cells (Extended Data Fig. 2) across rural domains reflect regional climate patterns. In general, fewer storms occur in the northern and western sectors, which are drier and farther away from the warm, moist Gulf Coast[27], whereas more frequent local-scale single-cell and isolated storms are observed in the wetter eastern sector. Tropical systems affecting these Texas cities typically originate from the Gulf Coast, resulting in a higher frequency in the southern rural areas. Warm-frontal and cold-frontal systems exhibit distinct spatial patterns, with more cold fronts in the north and more warm fronts in the east.

We compute the average storm counts across four rural domains and quantify deviations in urban storm occurrence relative to these rural averages. All four cities exhibit a higher frequency of isolated storms compared with their rural counterparts, with 7–16% increases in three inland cities. By contrast, Houston demonstrates a larger increase of 31%, owing to further sea–land interactions. A similar enhancement is observed for local-scale single-cell storms in most cities. We apply the non-parametric paired Wilcoxon signed-rank test to assess the statistical significance of urban–rural differences in storm counts (Supplementary Table 1). The results indicate that urban areas experience significantly more frequent local-scale single-cell storms ($P = 0.0059$, $r = 0.29$, $n = 92$) and isolated storms ($P = 0.0011$, $r = 0.34$, $n = 92$). By contrast, we do not find consistent urban effects on storm counts for storm types driven by larger-scale forcings, such as frontal storms and tropical systems. Further evidence of anthropogenic influence on local-scale storms is a weekend–weekday comparison (Extended Data Fig. 3). Weekends are found to have different temperature patterns and anthropogenic pollutant levels compared with weekdays[28–30], both of which can influence storm development. For single-cell storms and isolated storms, we find a decreased high-reflectivity grid cells (−6% to −36%) on weekends relative to weekdays across four cities, whereas no consistent weekend–weekday differences are observed for larger-scale storm types. A paired Wilcoxon signed-rank test confirms the significance for single-cell storms ($P = 0.0078$, $r = 0.28$, $n = 92$), whereas isolated storms show a weaker effect ($P = 0.067$, $r = 0.19$, $n = 92$). These findings are consistent with mechanisms identified in previous numerical studies, which show that urban heat islands enhance atmospheric instability and promote the development of local convection-driven storms in densely populated areas[10,11].

We further analyse the spatiotemporal variations in high-reflectivity grid cells ($\geq 40$ dBZ) from these two types of local-scale storm. Supplementary Fig. 1 presents histograms of high-reflectivity grid cell heights for all five storm types. At lower levels (1 km altitude), we observe more high-reflectivity grid cells over urban areas than over surrounding rural domains for all storm types and cities, except for San Antonio. This difference may reflect radar artefacts, such as ground blockage or surface clutter in urban environments[31,32]. To minimize potential radar errors near the surface, we exclude reflectivity data below 1 km

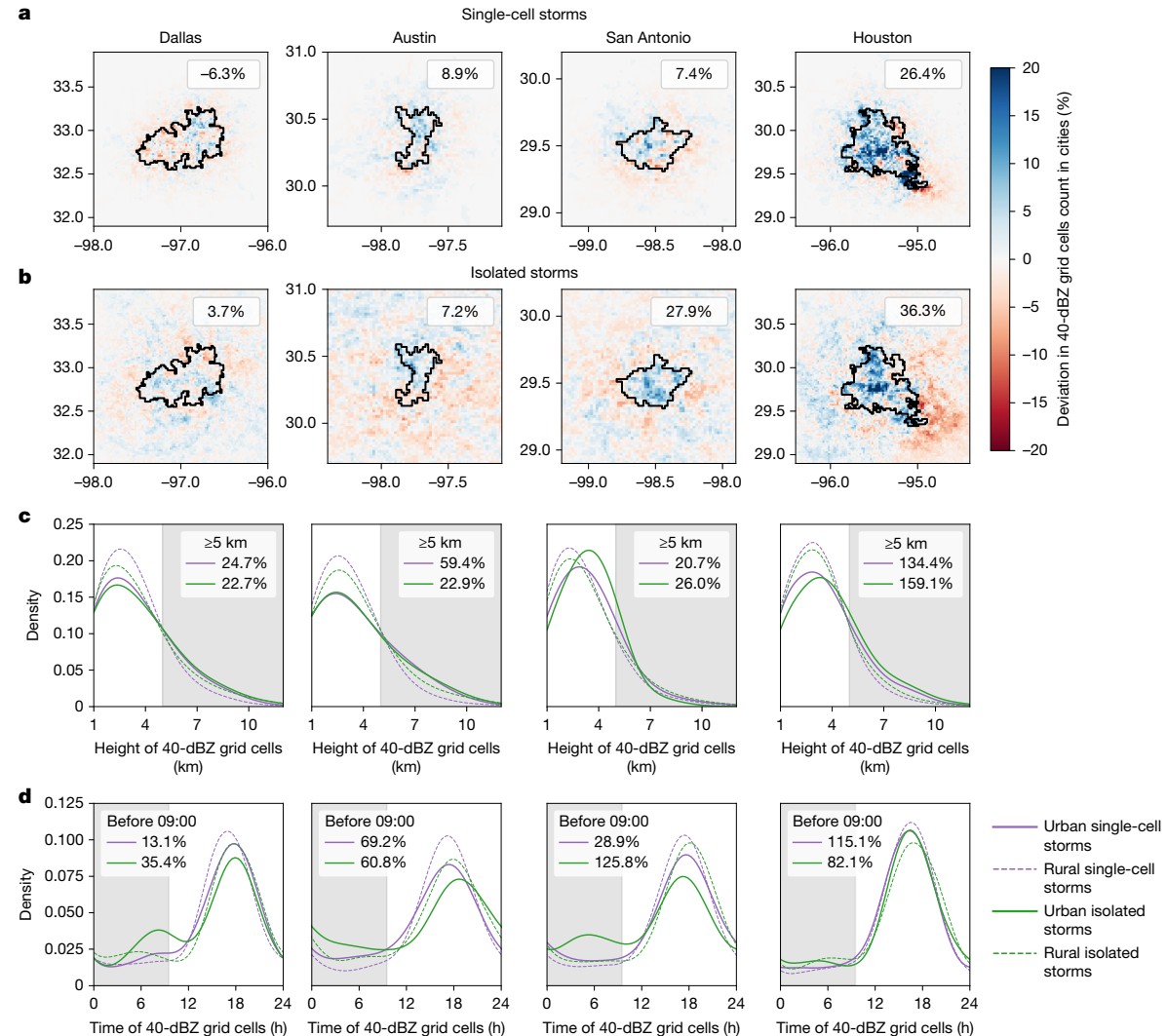

**Fig. 3 | Urban enhancement of local-scale single-cell and isolated storms.**
**a**,**b**, Spatial anomalies in the occurrence of high-reflectivity grid cells (≥40 dBZ, above 1 km) in urban areas relative to their rural counterparts. The black polygon marks the urban boundary, and the anomalies within this boundary are labelled in the panels. Paired Wilcoxon signed-rank tests confirm a significant increase in annual high-reflectivity occurrence for single-cell storms ($P = 0.0026$, $r = 0.31$, $n = 92$) and a marginal increase for isolated storms ($P = 0.056$, $r = 0.20$, $n = 92$) in urban areas. Spatial comparisons for the other three storm types are shown in Supplementary Fig. 2. **c**,**d**, Vertical (**c**) and diurnal (local time) (**d**) probability distributions of high-reflectivity grid cells for urban and rural storms. Urban anomalies above 5 km or before 09:00 local time (shaded regions) are quantified in the legend. For Houston, because the southern rural domain extends over the ocean, for which rainfall patterns differ from those over land, we exclude this domain from the rural comparison.

altitude from the following spatial analysis. Figure 3a,b shows the spatial anomalies in high-reflectivity grid cells for single-cell and isolated storms, respectively. Across all four Texas cities, urban areas exhibit increased high-reflectivity grid cells related to isolated storms (4–36%; Fig. 3b), consistent with their higher storm frequencies (7–31%; Fig. 2). The enhancement is more pronounced in southern cities, particularly Houston (36%) and San Antonio (28%). For single-cell storms, we find increased high-reflectivity grid cells in Houston (26.4%), Austin (8.9%) and San Antonio (7.4%) relative to their rural comparisons. By contrast, the Dallas–Fort Worth metropolitan area shows a slight decrease (−6.3%), although positive anomalies are shown in the eastern region, in which Dallas is located.

On the basis of previous results showing more frequent local-scale storms and increased high-reflectivity grid cells in urban areas, we further examine the vertical probability distributions to identify the primary layers of enhancement (Fig. 3c). In general, 80–90% of high-reflectivity grid cells occur below 5 km, with peaks around 2–3 km for both types of local-scale storm. However, when comparing urban and rural domains, most of the enhancement occurs aloft: above 5 km,

all cities show increases in high-reflectivity grid cells ranging from 21% to 59% in inland cities and 134% to 159% in Houston. This increase in upper-level reflectivity leads to a rise in average storm height: single-cell storms are elevated by 186–423 m in cities, whereas isolated storms show changes of −64 to 525 m with the lowest 1 km of the atmosphere excluded (Supplementary Fig. 1). Paired Wilcoxon signed-rank tests verify a significant increase in the annual mean height of high-reflectivity areas for both single-cell storms ($P < 0.001$, $r = 0.57$, $n = 92$) and isolated storms ($P = 0.004$, $r = 0.30$, $n = 92$). These results provide observational support for stronger and deeper convection over urban areas reported in previous numerical studies[11,33]. For other storm types, we do not observe consistent or significant changes in high-reflectivity height across four cities, except for tropical systems: we find a consistent, although not statistically significant, downward shift in high-reflectivity regions for tropical systems across four cities. Over rural areas, most high-reflectivity grid cells are concentrated at 2–3 km, whereas they shift downward to 1–2 km over urban areas (Supplementary Fig. 1c). The mean height of high-reflectivity grid cells decreases by 13 m in Houston and up to 304 m in Austin, even when the lowest 1 km is excluded. Although this shift is not

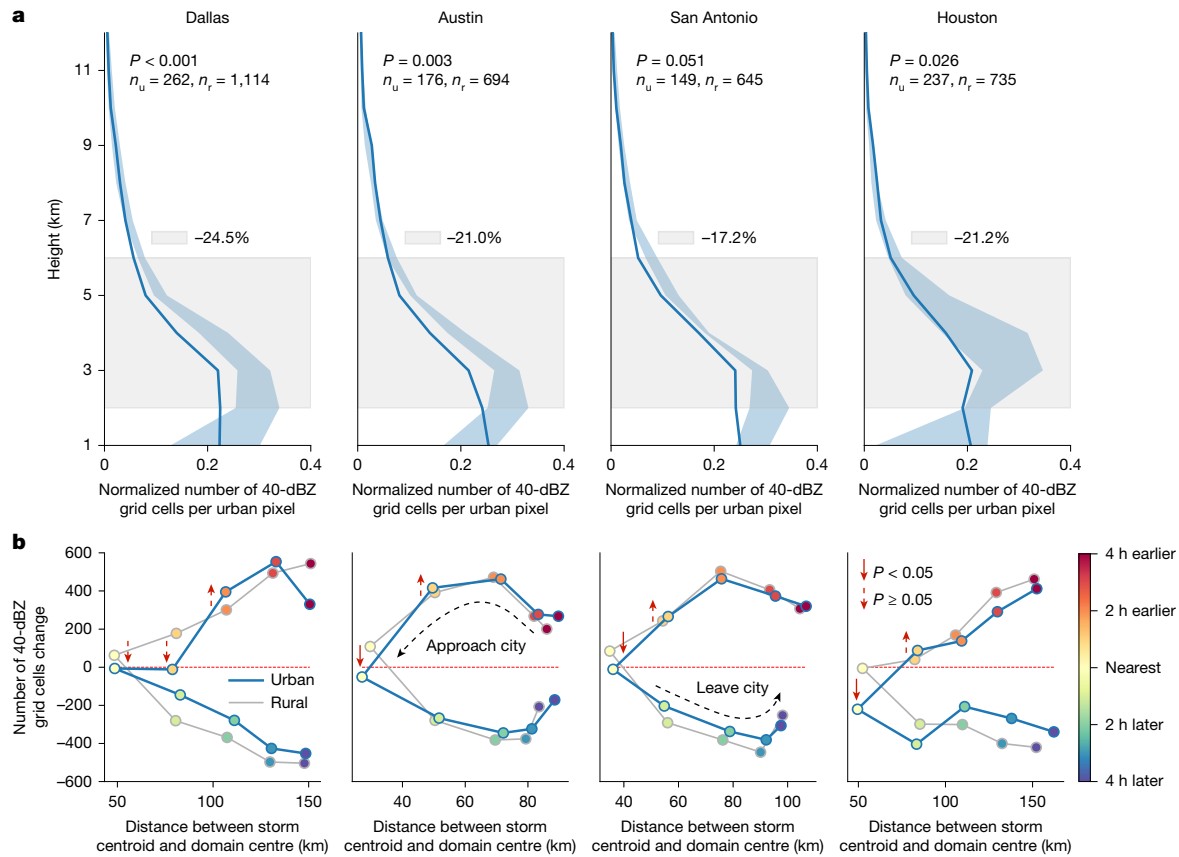

**Fig. 4 | Cold-frontal storm intensity weakens over urban areas compared with rural surroundings. a**, Vertical profiles of average cold-frontal storms passing over cities (solid lines) and four surrounding rural domains (blue-shaded areas represent the range across four rural domains). Significantly fewer high-reflectivity grid cells are observed in the lower to mid-troposphere (2–6 km) over urban areas compared with rural counterparts, with anomalies and Wilcoxon signed-rank test results indicated. The values of $n_u$ and $n_r$ denote the number of cold-frontal storms passing over urban and rural domains, respectively. **b**, Dynamical evolution of cold fronts approaching and moving away from urban and rural domains. When cold fronts approach cities, a larger fraction of cold-frontal storms enters the observation window, leading to an increase in high-reflectivity grid cells (above the zero line) and vice versa as storms move away. We find that cities may attenuate cold-front intensity at its nearest approach but slightly enhance intensity ahead of arrival. Non-parametric Wilcoxon signed-rank tests confirm significant decreases in high-reflectivity grid cells as cold fronts arrive in most cities (except Dallas, $P = 0.057$), whereas increases ahead of cities are not statistically significant.

statistically significant owing to the limited number of tropical systems in the 23-year record, it is noteworthy and warrants further investigation of tropical system structure and intensity using larger samples[12].

As well as spatial patterns, we examine the diurnal variation of high-reflectivity grid cells for these local-scale storms. Single-cell and isolated storms demonstrate a more pronounced diurnal cycle than larger-scale storms, with peaks in the late afternoon around 18:00 local time. This finding reinforces the role of surface heating in driving convection. Comparing urban and rural distributions, we find that increases in high reflectivity occur throughout the day; however, the relative differences reveal a distinct urban precipitation enhancement at night (Fig. 3d). Before 09:00 local time, urban areas have 13–126% more high-reflectivity grid cells than rural areas. This nocturnal enhancement highlights the role of urban heat islands in modulating night-time urban boundary layer[34].

## Opposite urban effects on frontal storms

Although the number of cold-front or warm-front events does not differ significantly between urban and rural areas, cold-frontal storms produce significantly fewer high-reflectivity grid cells (≥40 dBZ) over cities (−16% to −28%; Supplementary Fig. 2). In other words, urbanization may not affect the frequency of frontal storm events, as it does for local-scale storms, but it can influence their intensity. We observe more than 140 cold-frontal storms in each urban and rural domain over 23-year warm seasons ($n_u$ and $n_r$ in Fig. 4a), which makes a composite analysis possible: averaging the high-reflectivity grid cells of cold-frontal storms (Fig. 4a), we find weaker cold-frontal storm intensity over urban areas, with a 17–24% decrease in high-reflectivity grid cells between 2 km and 6 km altitude. The diurnal histogram (Extended Data Fig. 4) indicates that this decrease can occur at any time of the day. This figure also shows an interesting diurnal feature of cold-frontal and warm-frontal storms that are not directed to urban landscapes: these storms exhibit lower reflectivity around noon in inland regions, in contrast to the afternoon peak seen in local convective storms. The weakening of frontal storms during the midday period could be related to surface heating, which enhances boundary-layer mixing and drying, or to a reduction in the temperature contrast across the front. Such land–atmosphere interactions have not been adequately captured in previous observational or numerical studies and deserve more investigation in the future.

To investigate the dynamic evolution of cold fronts approaching and moving away from cities, we analyse the changes in cold-front intensity along their trajectories. We calculate the hourly storm centroid locations, their distances from the city centre and the corresponding variations in high-reflectivity grid cells within the observation window and then plot the trajectories for urban and rural cold-frontal storms (Fig. 4b). The number of high-reflectivity grid cells increases as cold fronts approach the domain and decreases as they arrive.

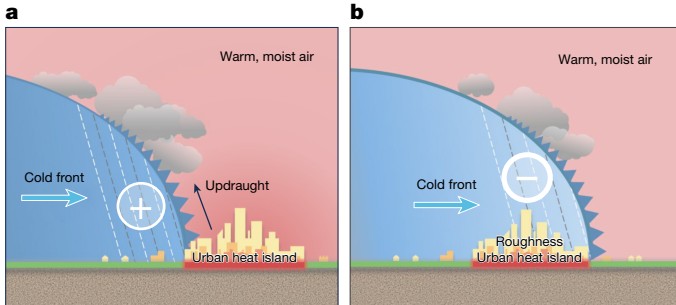

**Fig. 5 | The dual influence of urbanization on cold-frontal storm intensity. a**, As a cold front reaches the city boundary, the urban heat island may enhance the thermal gradient across the front, potentially intensifying storm activity. **b**, As the cold front moves into and sweeps over the city, storm intensity decreased significantly, probably because of a weakened thermal gradient and increased surface roughness, which could disrupt the lower part of the frontal system.

Comparing urban and rural trajectories, we find a consistent weakening of cold-front intensity near cities. Specifically, reflectivity decreases significantly at its nearest approach ($P < 0.05$ for most cities; $P = 0.057$ for Dallas), whereas slightly enhanced reflectivity is observed just before reaching urban areas. We further examine spatial composites of cold-frontal storms at their nearest approach to the cities, as well as 1 h before and after, in Extended Data Fig. 5 and Supplementary Figs. 3–5. These maps show that the cold-frontal storms in Texas generally move from northwest to southeast at speeds of 40–50 km per hour. When the storms are at their closest approach to the city centre, all four cities show varying degrees of reduction in the high-reflectivity grid cells, with some areas showing a 10% reduction. One hour earlier, we observe an approximately 9% increase in the probability of high-reflectivity grid cells northwest of Austin (Extended Data Fig. 5) and San Antonio (Supplementary Fig. 3).

The observed decrease in cold-frontal storm intensity over urban areas, along with the potential enhancement ahead to the city, can be explained by the thermal-driven mechanism of frontal storms. Most cold-frontal storms are driven by cold air masses or gust fronts advancing into warmer environments. As illustrated in Fig. 5, before a cold front reaches a city, the urban heat island effect creates a warmer environment ahead of the front, increasing the temperature gradient across the front and enhancing storm intensity on the upward (northwest) side of the city. However, as the cold front moves into and sweeps over cities, the cold air mass is altered by urban heat and increased surface roughness, which can disrupt the lower part of the frontal system and lead to a significant decrease in storm intensity. Gaffen and Bornstein[35] documented how urban roughness can disrupt a cold front during its passage over New York City. Hu et al.[36] reported a similar intensification–attenuation pattern for cold fronts over the Greater Bay Area[36]. Their numerical simulations attributed these variations in frontal intensity to changes in the meridional gradient of equivalent potential temperature at 925 hPa.

We find different urban effects on warm-frontal storms relative to cold fronts. Specifically, we do not detect an increased number of warm-front events or more high-reflectivity grid cells over urban areas compared with rural areas. However, when investigating reflectivity values above 40 dBZ and their vertical distribution, all four cities show higher mean reflectivity in urban areas, although these differences are not statistically significant. Extended Data Fig. 6 compares the average high reflectivity (≥40 dBZ) for individual warm-frontal storms passing over urban and rural domains. Most quantile–quantile curves lie above the 1:1 line, indicating stronger storm intensities over urban areas. Also, the mean height of these high-reflectivity grid cells increases by 169–260 m in three cities, whereas in San Antonio it is, on average, 39 m lower. Because western San Antonio is characterized by a hilly terrain, we exclude the western rural domain and find that the urban mean height becomes 21 m higher. Although these differences in reflectivity and height do not lead to statistically significant results, they suggest that different urban influences on cold-frontal and warm-frontal storms. Warm-frontal storms seem to be less sensitive to urbanization, yet may still exhibit modest intensification. The underlying mechanisms may be opposite to those affecting cold fronts: urban heat islands may enhance the warm sector of warm-frontal storms and promote storm development, whereas stronger updraughts further elevate storms. Previous numerical simulations have mainly focused on other storm types, leaving the mechanisms governing warm-front responses largely unexplored. Because warm-frontal precipitation is typically associated with stratiform clouds and weaker vertical motion, it is expected to be less responsive to surface thermal and dynamical perturbations[37]. These findings call for further observational and modelling investigations to better understand how urbanization influences warm-frontal storms.

## Discussion

Our previous work examined long-term climatological changes in urban precipitation and found inconsistent results across global[3] and US cities[26]. This study helps reconcile those inconsistencies by explicitly accounting for storm dynamics and vertical structures. By developing an objective storm identification and classification algorithm, we establish a framework for event-based analysis of storm behaviour. We show that the effects of urbanization are not uniform but vary with storm mechanisms and scales. Local-scale single-cell storms and isolated storms occur more frequently over cities, particularly at night, consistent with previous numerical studies demonstrating urban-heat-island-driven atmospheric instability[9–11]. For frontal storms, urbanization mainly influences intensity rather than frequency: cold fronts are significantly weakened over cities, probably because increased surface roughness and urban heating disrupt their dynamical structure and thermal gradients, whereas warm fronts show moderate but non-significant intensification. Numerical simulations for New York City and the Greater Bay Area attribute these changes in cold-frontal intensity to both dynamical and thermal effects[35,36], whereas few studies have documented any urban-induced changes in warm-frontal storms. This contrast probably reflects fundamental differences in storm structure and mechanism: cold-frontal storms are typically narrow, deep convective systems, whereas warm-frontal storms are dominated by broad stratiform clouds, making the latter less sensitive to land-surface perturbations[37]. Tropical systems show no systematic changes in frequency or intensity but shift high-reflectivity regions at lower altitudes over urban areas, where reflectivity is more directly linked to surface rainfall than aloft. This observational finding, derived from three-dimensional radar reflectivity data, contrasts with previous modelling results that reported urban-induced intensification[12,13]. This discrepancy may arise from differences between observational reflectivity analysis and modelling approaches to precipitation intensity or uncertainties associated with the limited sample size within the 23-year record in Texas. By capturing storm types ranging from convective to frontal and tropical systems, this study helps reconcile the distinct urban effects reported in previous studies[14,15]. For example, Chang et al.[38] distinguished between short-duration and long-duration rainfall events and found enhanced short-duration rainfall but suppressed long-duration rainfall over cities. Our results support and extend these findings by linking short-term rainfall to local-scale convective storms and long-term rainfall to synoptic frontal systems.

This study focuses on four Texas cities that encompass diverse climatic and geographic settings. Dallas and Houston are relatively flat, whereas Austin and San Antonio have more pronounced topographic

variation. Houston is coastal, whereas the other three are inland. Despite these differences, we identify broadly consistent urban storm effects across cities, suggesting robust city–storm interactions. Nonetheless, Texas does not encompass the full range of global climates. Although we do not explicitly address broader regional climate forcings, our findings, together with previous research[26], suggest that urbanization effects persist alongside stronger regional drivers rather than disappearing. An example from this study is Houston. Proximity to the warm Gulf Coast weakens cold-frontal systems, making urban weakening effect less pronounced than in inland cities. Also, this study focuses on the warm season and does not include other important systems, such as orographic rainfall, monsoon storms[39] or mesoscale convective systems. Future studies are encouraged to extend this framework to a broader range of weather systems and regions. This research advances our understanding of city–storm interactions and underscores the importance of incorporating storm-type specificity into Earth system and atmospheric models to improve urban rainfall simulations. It further implies that infrastructure design standards based on aggregated rainfall statistics may fail to capture critical extremes associated with specific storm processes, highlighting the importance of integrating storm-type information into urban hydrological design, early-warning systems, and resilience planning.

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

## Methods

### Radar reflectivity data

To characterize the three-dimensional structure of storms, weather radar provides the most suitable observational data. Unlike conventional satellite or rain-gauge observations, which offer precipitation intensity on a two-dimensional spatial scale, radar measures the strength of signals reflected to the receiver at different angles, thereby preserving the vertical structure of storms. These reflected signals, known as reflectivity, represent a combination of raindrop size and number concentration and are therefore related to precipitation intensity. Reflectivity is measured as $Z$ ($mm^6 m^{-3}$) and is typically expressed on a logarithmic scale as decibels of reflectivity (dBZ). This study analyses reflectivity directly rather than converting it to rain rate ($R$), thereby avoiding uncertainties in the $Z–R$ relationship, which is a main source of error in radar-based precipitation retrievals.

This research uses GridRad data version 3.1, which is specifically designed to investigate deep convective systems and evaluate the vertical structure of storms[40,41]. This dataset provides hourly, three-dimensional reflectivity (0.02° latitude × 0.02° longitude × 1 km altitude) across the contiguous United States from 1995 to 2017, determining the time frame of this study[25]. GridRad data have been widely used to analyse mesoscale convective systems[42,43], hail[44] and tropopause-overshooting convection[45]. It merges reflectivity observation from 125 National Weather Service NEXRAD WSR-88D Level II weather radars[46] onto a common three-dimensional grid through a four-step algorithmic procedure: (1) reading raw data; (2) identifying grid volumes; (3) computing space–time weights; and (4) applying weighted binning[40]. As well as the space–time weighting scheme that reduces noise from low-quality or distant data, GridRad applies several quality control procedures, such as filtering low-confidence echoes and removing ground clutter, to minimize non-meteorological artefacts. Details of these quality control procedures and validation against other radar datasets are described in the algorithm description documentation[40]. To further reduce residual noise caused by remaining artefacts, this research applies two reflectivity thresholds to identify storms: 20 dBZ to differentiate rain from drizzle, very light rain or non-precipitating echoes, and 40 dBZ (approximately 10 mm h$^{-1}$ rain rate) to identify heavy rainfall[47,48]. The identified storms with areas smaller than 100 km$^2$ are excluded to further reduce uncertainties associated with small-scale artefacts.

### Research domains

This research focuses on four main cities in Texas: Dallas–Fort Worth, Austin, San Antonio and Houston. We first define urban areas based on the 2019 National Land Cover Database (NLCD)[49]. To provide a basis for comparison, we set up four rural domains to the north, south, west and east of each city as rural comparisons. In total, we define 20 research domains: one urban and four rural domains for each of the four cities (Extended Data Fig. 1). To determine the locations of the four rural comparison domains, we calculate the spatial extent of city development along the longitude and latitude directions, the average of which is defined as the urban diameter. Because urban influences on precipitation extend beyond city boundaries, we test several distances of 0.5, 1.0 and 1.5 times the urban diameter from the urban domain as rural domains. Distances that are too small may result in overlapping urban influences and reduce contrast between urban and rural storms, whereas larger distances may lead to proximity to other cities owing to the dense distribution of Texas urban areas. We finally select the rural comparison domains as areas one urban diameter beyond each urban domain. For Houston, which is located along the Gulf Coast and has a large urban extent, further adjustments are made. Specially, we move the eastern rural domain northward by 0.5 urban diameters to avoid ocean coverage and move the western rural domain eastward by 0.5 urban diameters to avoid overlap with the Austin region. The final

domain configuration is shown in Extended Data Fig. 1. To capture the dynamics of storm evolution approaching and moving away, we define an extended observation window for each domain by adding a further urban diameter in all directions.

### Storm identification

We extract individual storm events from the hourly three-dimensional gridded reflectivity data based on two different thresholds (20 dBZ and 40 dBZ). Our method began by locating high-reflectivity grid cells above 40 dBZ in the 20 research domains during the warm season (May–September) across 23 years. These high-reflectivity grid cells serve as starting points, from which we expand to all adjacent rainy grid cells above 20 dBZ in a three-dimensional volume to define the full rainy area. Finally, to capture the entire storm event from the time series, we track overlapping rainy areas across consecutive hourly time intervals. Those overlapping rainy areas are combined into a single storm event. Therefore, distinct and unconnected convective cells are identified as individual storm events. As a result, it is possible to observe several storm events (typically fewer than three) occurring simultaneously within a given domain.

In this way, we preliminarily identify approximately 1,000 to 12,000 storm events across each of the 20 research domains. Through the visual inspection of radar animations, we identify several issues, prompting us to enhance the algorithm with further steps for correction. First, we find that isolated spikes in radar reflectivity data are occasionally misidentified as small storm events. To reduce such artefacts, we exclude storms with a peak rainy area smaller than 100 km$^2$. Second, some distinct storms may be erroneously merged into a single event owing to overlapping rainfall areas. For example, a warm front followed by a cold front may be mistakenly classified as a single storm event. To resolve this, we separate time series containing several periods of high reflectivity into distinct storm events. For example, a storm event with a time sequence of maximum reflectivity as 21, 30, 42, 42, 25, 21, 45, 30 and 26 dBZ will be divided into two separate events, with the split occurring at the time step corresponding to the minimum number of rain grid cells (≥20 dBZ) between peaks. Finally, some storms intersect marginally with research domains, with most of the areas outside the area of interest. To address this, we track the storm motion by calculating the reflectivity-weighted centroid at each time step (weights are defined as reflectivity values above the 40-dBZ threshold). If the storm centroid trajectory does not intersect with the study scope (including a 0.05° buffer zone) and the high-reflectivity area covers less than 20% of the domain, we presume limited urban influences and excluded those storms from analyses. After these refinements, the final dataset includes approximately 1,000–5,000 storm events per domain. A flowchart illustrating storm identification and post-processing procedures is shown in Supplementary Fig. 6.

### Storm classification

We classify the identified warm-season storm events into five types: single-cell storms, isolated storms, tropical systems and warm-frontal and cold-frontal storms. Similar to the storm-identification process, we first examine radar animations of numerous storms to subjectively characterize the features of each storm type in Texas. For instance, we observe that single-cell storms and isolated storms are usually localized and short-lived compared with other storms; frontal storms tend to persist longer with warm and cold fronts moving in different directions; tropical systems generally have longer duration and larger rainfall areas, and their large spatial scale causes storm centroids to move more slowly than frontal storms. We then translate these observed storm-performance characteristics into objective criteria based on storm properties, including rainy area, heavy rain area, duration, movement speed and direction: the rainy area and heavy rain area are quantified as the largest two-dimensional contiguous regions with the column-maximum reflectivity exceeding 20 dBZ and 40 dBZ,

respectively, throughout the storm passing through the observation window; storm speed and direction are calculated from the movement of storm centroid, in which the hourly centroid displacements are computed, and their median values are used.

By visually inspecting the radar animations, we roughly estimate the initial maximum and minimum thresholds to determine a parameter uncertainty range. For storms falling within this range, we further examine synoptic conditions and assign storm types by referencing the National Oceanic and Atmospheric Administration (NOAA) weather archive of the national forecast charts[50], wind conditions and temperature changes before and after each event. Using this labelled subset, we calibrate the final thresholds for parameters and refine the storm-classification scheme (Extended Data Table 2). This yields an objective storm-classification method, which is then applied to all storm events. The classification results for 940 warm-season storms in Austin are illustrated in Supplementary Fig. 7 as an example. Animations of representative storms across different types are available from Zenodo (https://doi.org/10.5281/zenodo.15933280) with corresponding hourly snapshots provided in Supplementary Figs. 8 and 9.

### Sensitivity and uncertainty analyses

Limitations of this study arise mainly from uncertainties in the storm identification and classification methods. The objective classification algorithm developed here builds on the earlier subjective classification approach of Lorenz et al.[47]. As well as the rainfall and heavy rainfall areas considered in that work, our method incorporates dynamic characteristics, including storm movement and duration. Given the large sample size, we do not examine synoptic conditions for every case, such as temperature gradients or sustained winds. Instead, we only investigate synoptic conditions for a limited subset of storms near the classification boundary to calibrate the objective criteria thresholds. This compromise between classification accuracy and computational feasibility enables us to move from subjective classification to a fully automated, objective framework applicable to a large number of storm events. Strictly speaking, this algorithm is based on storm behaviour and dynamic properties rather than predefined storm type categories, although it aligns well with commonly recognized storm types. Rather than strictly identifying meteorological storm types, we group storm events into five basic categories for urban–rural comparison. This simplification may introduce some uncertainty and mixing across more specialized storm types. We expect this approach to be well-suited for studying storm–urban interactions, as it relies on intrinsic storm properties that govern how storms respond to urban environments.

To assess the robustness of storm identification and the corresponding urban effects, we conducted sensitivity tests using different radar reflectivity thresholds. Because the 20-dBZ value corresponds to very light precipitation, lower reflectivity levels are unlikely to affect the main conclusions; therefore, we use elevated thresholds of 25 and 45 dBZ to repeat the analysis. Storm counts in Supplementary Table 2 show that local-scale single-cell and isolated storms are more sensitive to threshold selection, with approximately 40% and 20% fewer storm events, respectively. By contrast, larger-scale frontal storms and tropical systems exhibit greater robustness to threshold changes. Despite the reduced sample size, the urban–rural comparison of storm counts remains significant for single-cell and isolated storms (Supplementary Table 2). However, because these local-scale storms may be driven by the same mesoscale or synoptic-scale systems and exhibit spatial correlation, this could reduce the effective degrees of freedom in the paired city–year sample, introducing uncertainty into the significance test. For cold-frontal storms, the identified significant decrease in intensity over urban areas also remains robust (Supplementary Fig. 10). We further evaluated 26 tropically influenced weather systems identified in Houston using National Hurricane Center tropical cyclone reports, the Texas Climate Report and further sources (Supplementary Table 3). Among these 26 events, 14 were officially named hurricanes or tropical storms, whereas eight were other weather systems influenced by tropical air flow, including tropical waves, tropical-influenced convective storms and mixed weather systems that perform similarly to tropical storms within our observation window. This outcome is expected, as our classification emphasizes long event durations and large, heavy-rainfall footprints, which are typically associated with tropical moisture. Overall, the results indicate that our algorithm could reliably identify storms influenced by tropical systems. The classification criteria and specific thresholds used in our study are calibrated on the basis of storm characteristics within our observation windows. These thresholds should be recalibrated when applying this approach to other regions, particularly for identifying large frontal and tropical systems. Such adaptations should account for regional differences in storm pathways, typical trajectories, spatiotemporal scales and local climatological characteristics. Further details are provided in the Supplementary Information and rely on refs. 51–55.

## Data availability

The GridRad data version 3.1 are available at https://gridrad.org/index.html. The 2019 National Land Cover Database (NLCD) is available at https://doi.org/10.5066/P9KZCM54.

## Code availability

Code used for analysis and figure generation is available from Zenodo (https://doi.org/10.5281/zenodo.19339621)[56]. Animations of representative storm events are available in an earlier version (https://doi.org/10.5281/zenodo.15933280)[57].

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

**Acknowledgements** X.S. is supported by the Future Investigators in NASA Earth and Space Science and Technology (FINESST) Fellowship (80NSSC22K1540). This work constitutes the

final chapter of the PhD dissertation of X.S at the University of Texas at Austin. D.N. acknowledges support from the National Science Foundation (2413827), NIST (60NANB24D235), and NASA Interdisciplinary Science (IDS) grants (80NSSC20K1262 and 80NSSC20K1268).

**Author contributions** The authors confirm contributions to the paper as follows: research conception and design: X.S., J.N.-G., Z.-L.Y. and D.N.; data collection: X.S.; analysis and interpretation of results: X.S. and J.N.-G.; visualization: X.S.; draft manuscript preparation: X.S.; funding acquisition: X.S., Z.-L.Y. and D.N.; supervision: Z.-L.Y. and D.N. All authors reviewed the results and approved the final version of the manuscript.

**Competing interests** The authors declare no competing interests.

**Additional information**
**Correspondence and requests for materials** should be addressed to Xinxin Sui.

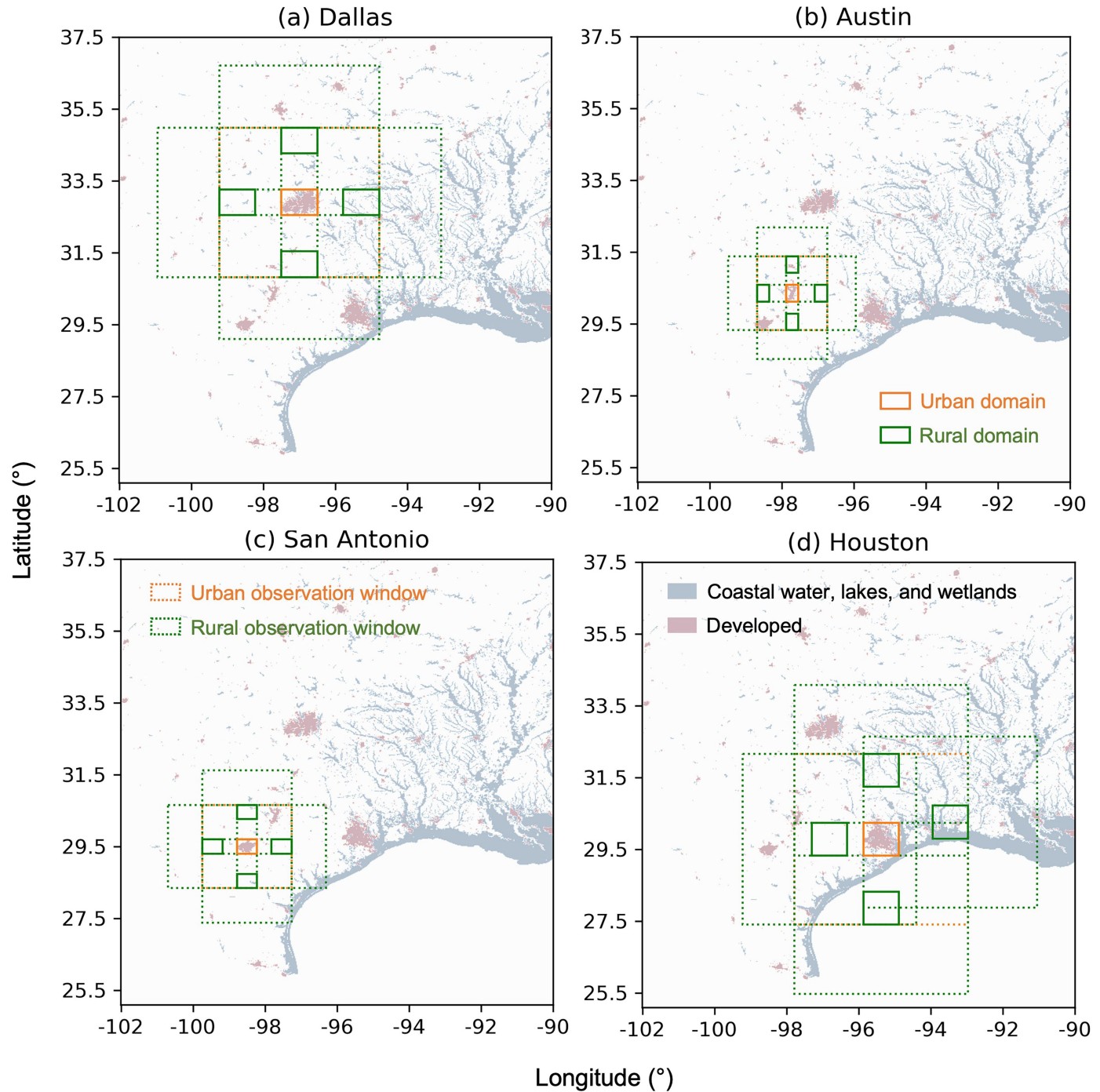

**Extended Data Fig. 1 | Urban and rural domains and the corresponding storm-observation windows.** The observation windows extend the urban footprint two times out of the city boundary. Land-cover data are derived from the National Land Cover Database (NLCD, 2019).

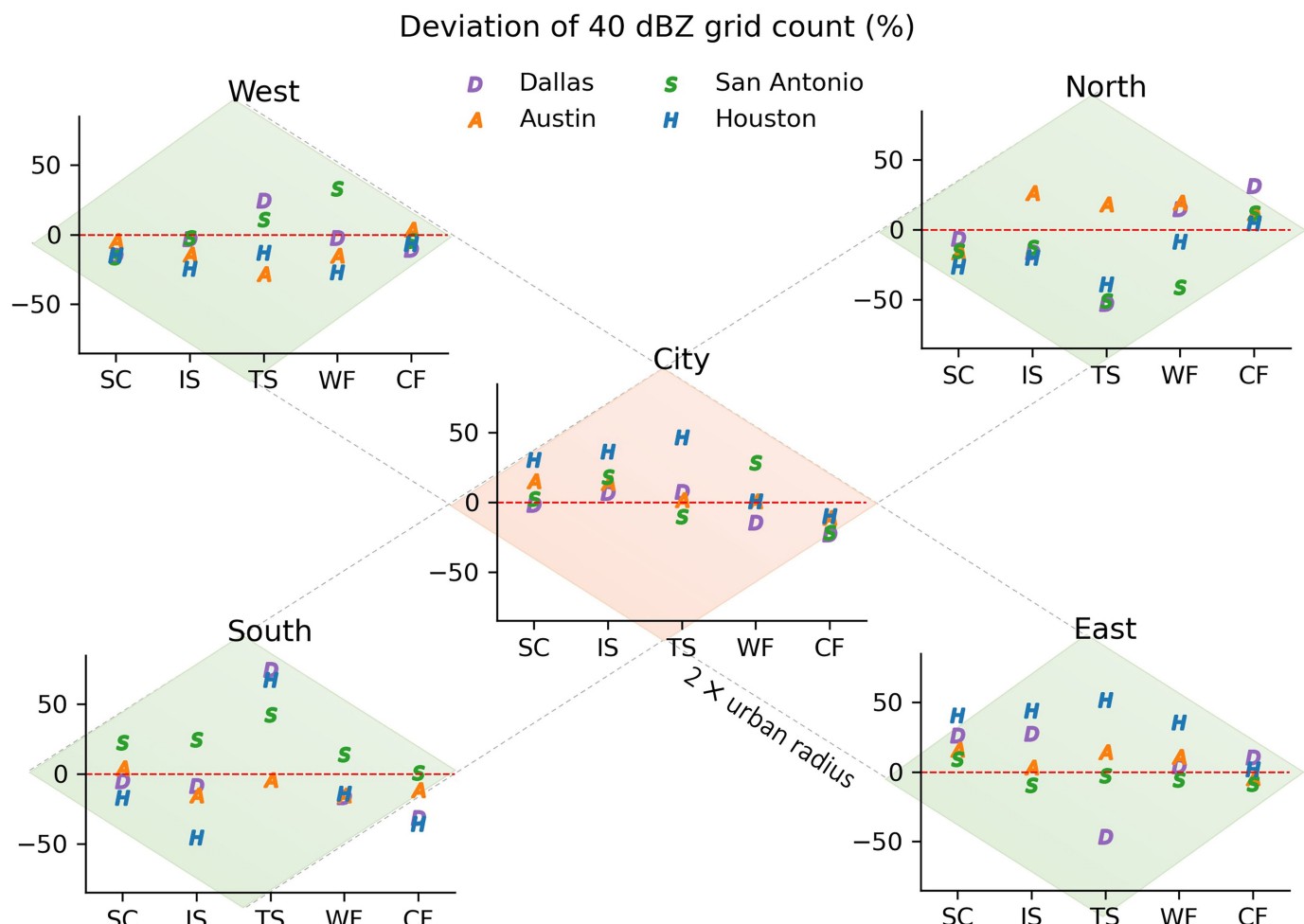

**Extended Data Fig. 2 | Urban–rural differences in high-reflectivity grid cells across storm types.** Deviation in the number of high-reflectivity grid cells (≥40 dBZ) for five types of warm-season storm in four Texas cities compared with the average cell number in their peripheral rural control areas (1995–2017). For Houston, as the south rural domain lies over the ocean, for which rainfall patterns differ from those on land, we exclude the data from this south rural area when calculating the rural baseline.

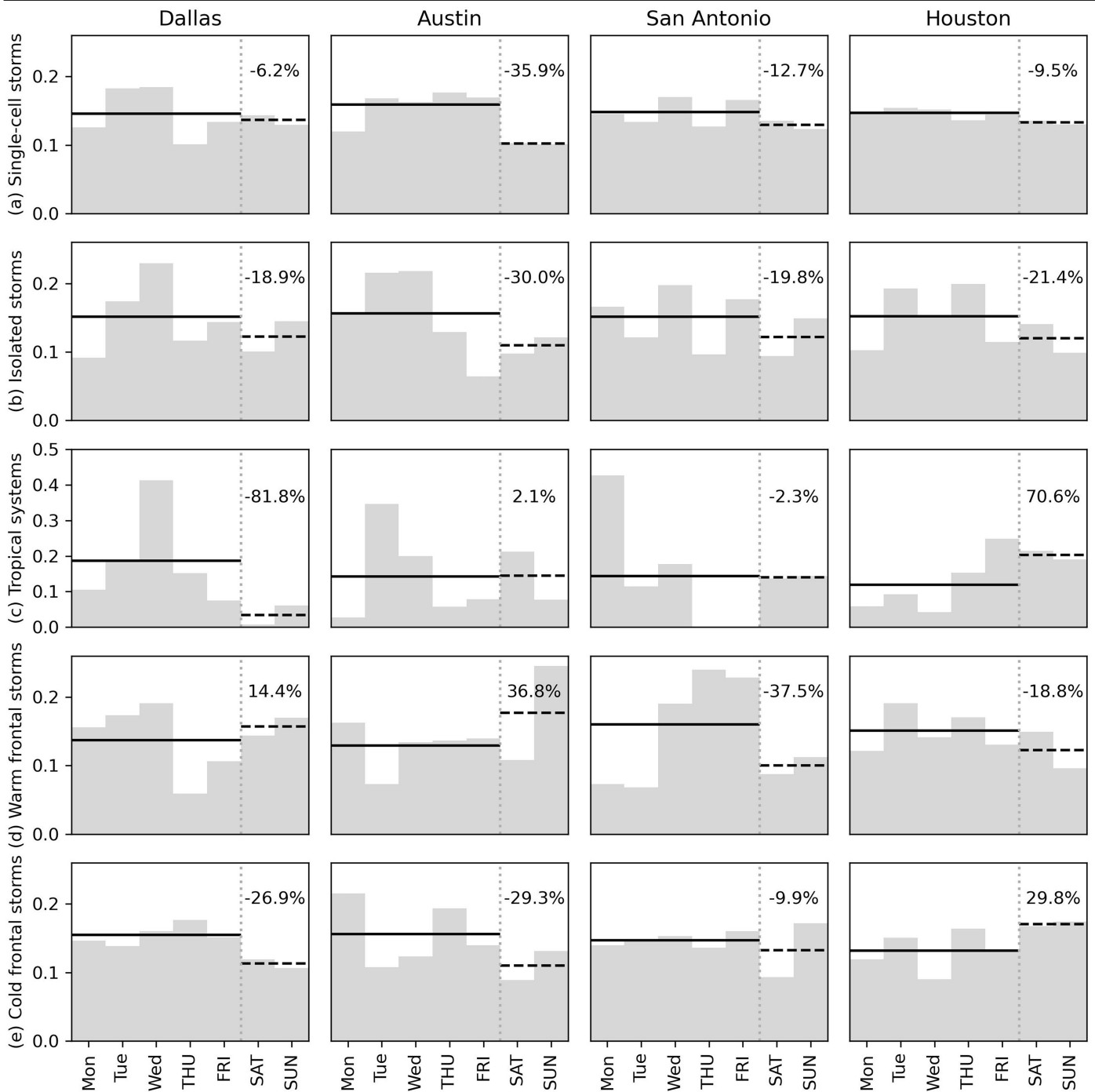

**Extended Data Fig. 3 | Weekday–weekend differences in high-reflectivity grid proportion are consistent for convective storms.** The solid lines represent the average density on weekdays and the dashed lines indicate the density on weekends, with weekend anomalies labelled. Paired Wilcoxon signed-rank tests report a significant increase in yearly high-reflectivity grid cells for single-cell storms ($P = 0.0078$, $r = 0.28$, $n = 92$) and a weaker increase for isolated storms ($P = 0.067$, $r = 0.19$, $n = 92$) on weekdays compared with weekends.

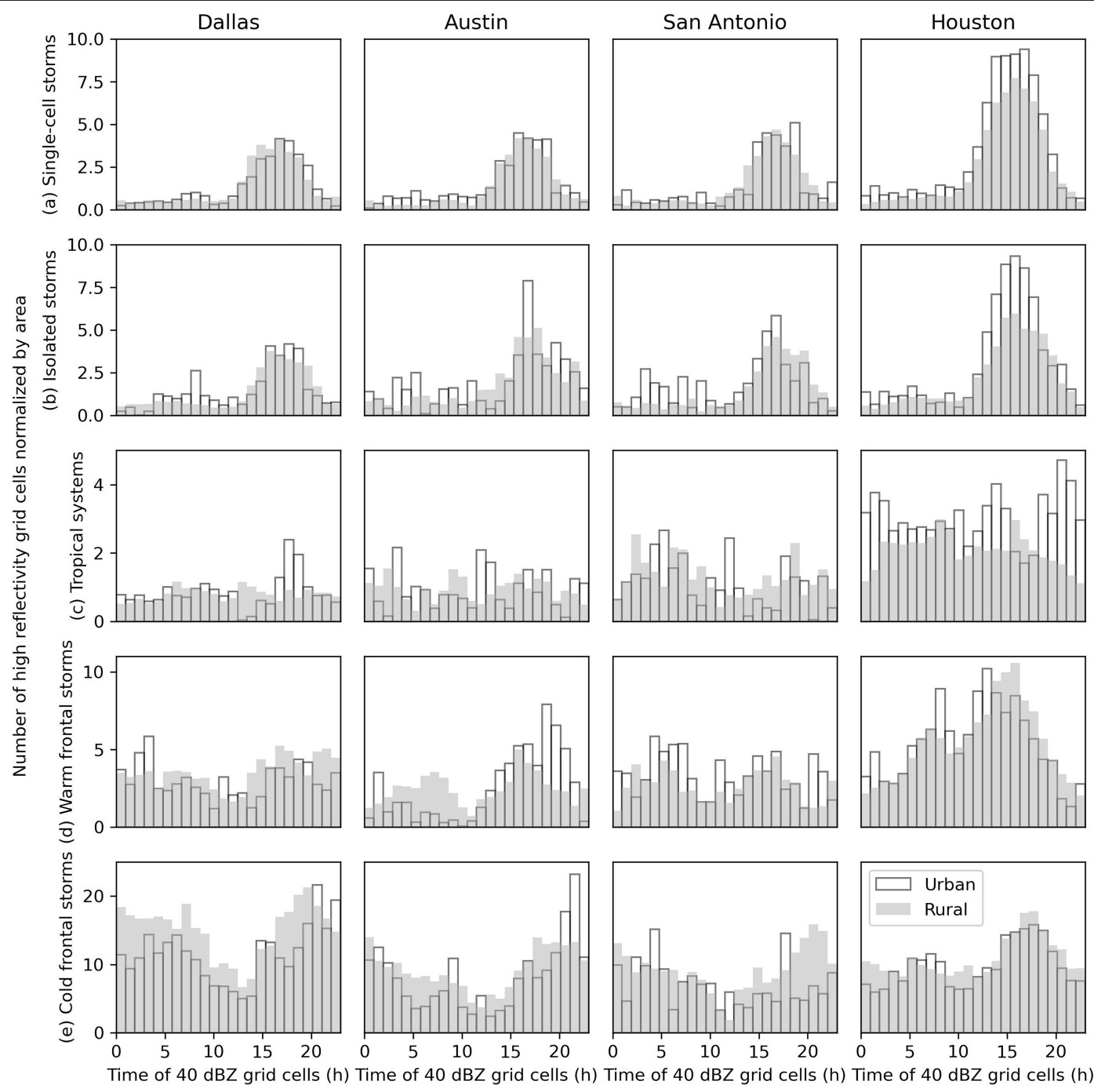

**Extended Data Fig. 4 | Histogram of the diurnal distribution of high-reflectivity grid cells in urban and rural domains.** Time is shown in local time.

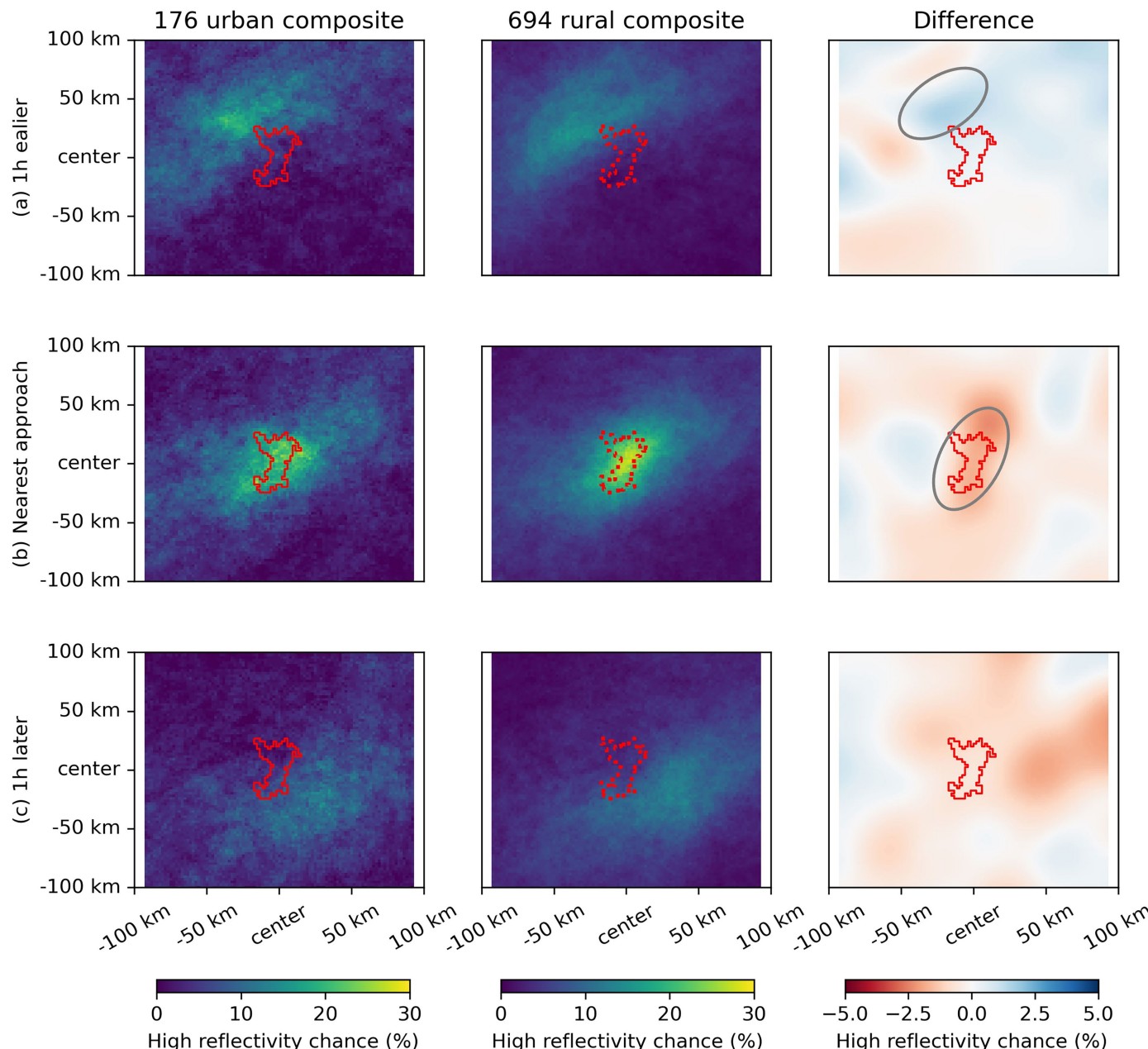

**Extended Data Fig. 5 | Cold-frontal storms weaken over Austin during passage and show enhancement ahead of arrival. a–c**, The spatial map shows the frequency of high-reflectivity grid cells one hour before (**a**), on arrival (**b**) and after arrival (**c**) in urban or rural domains, with cold fronts moving from the northwest towards the southeast. The difference between urban and rural cold-front composite is shown after applying a two-dimensional Gaussian smoothing filter. It indicates a significant reduction in reflectivity intensity over Austin on arrival time and a non-significant increase in high-reflectivity grid cells northwest of Austin 1 h before. The red polygon marks the Austin urban boundary within the observation window for reference.

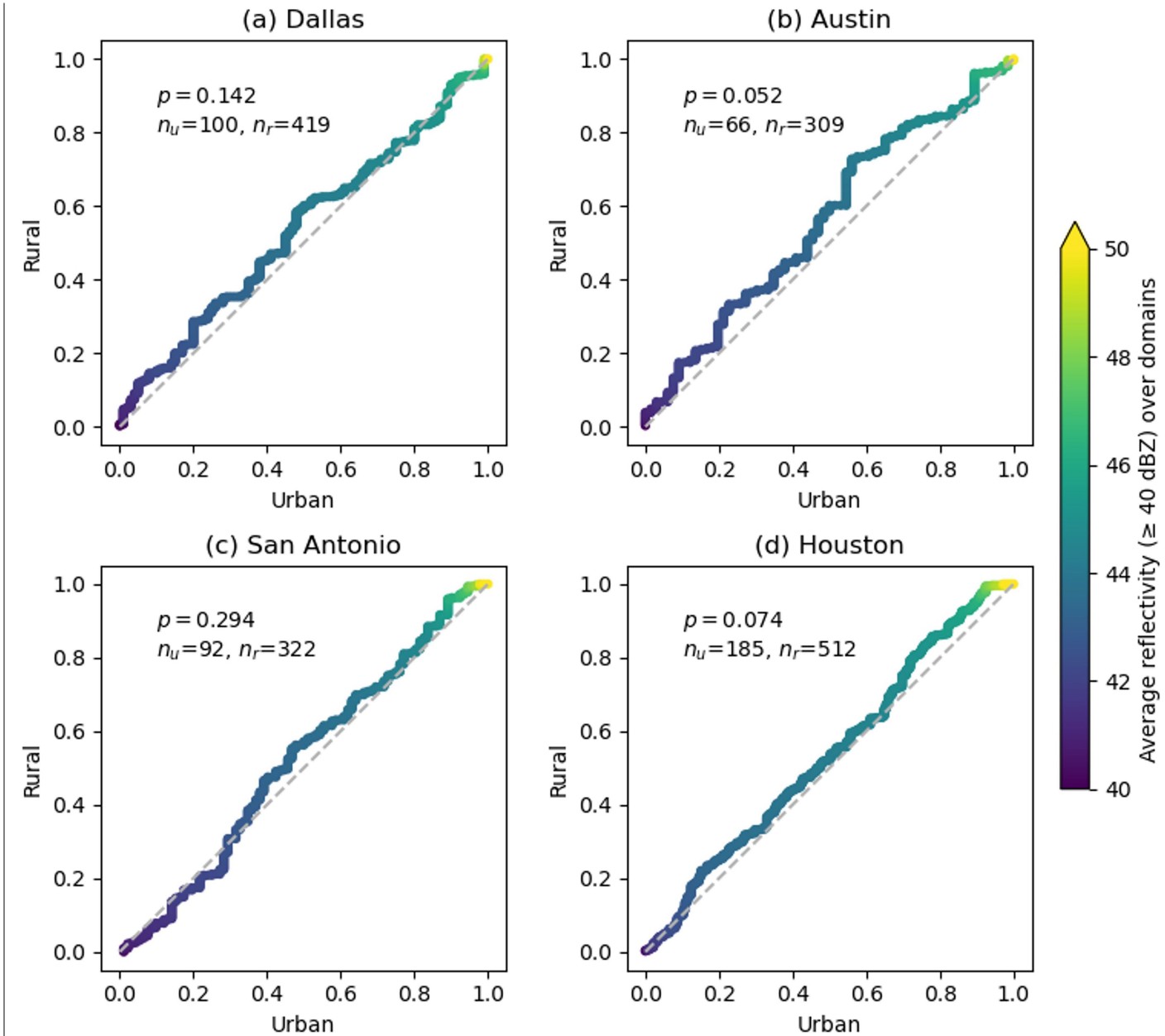

**Extended Data Fig. 6 | Quantile–quantile comparisons between urban and rural warm-frontal storm intensities.** Each point represents the average reflectivity (≥40 dBZ) of an individual storm over urban or rural domains. Most of the four quantile–quantile lines lie above the 1:1 line, indicating slightly stronger warm-frontal storm intensities over urban areas, although not statistically significant. The $n_u$ and $n_r$ values indicate the number of warm-frontal storms passing urban and rural domains, respectively.

**Extended Data Table 1 | Statistics of storm numbers in four Texas cities from 1995 to 2017**

|  | Dallas | Austin | San Antonio | Houston |
|---|---|---|---|---|
| Single-cell storm | 1,459 (70.1%) | 522 (55.5%) | 563 (59.1%) | 4,054 (81.2%) |
| Isolated storm | 247 (11.9%) | 158 (16.8%) | 138 (14.5%) | 491 (9.8%) |
| Tropical system | 11 (0.5%) | 16 (1.7%) | 9 (0.9%) | 26 (0.5%) |
| Warm-frontal storm | 102 (4.9%) | 68 (7.2%) | 93 (9.8%) | 185 (3.7%) |
| Cold-frontal storm | 262 (12.6%) | 176 (18.7%) | 149 (15.7%) | 237 (4.7%) |
| All | 2,081 | 940 | 952 | 4,993 |

**Extended Data Table 2 | Criteria for classifying five storm types for the four cities in Texas**

| Classification order | Storm type | Definition | Character |
|---|---|---|---|
| 1 | Single-cell storm | $A_R < 1{,}300\,\text{km}^2$, duration $\leq 5\,\text{h}$ | Small rain area, short duration |
| 2 | Isolated storm | $A_R < 7{,}800\,\text{km}^2$ or $A_{HR} < 520\,\text{km}^2$ | Relatively small rain and heavy rain area |
| 3 | Tropical system | Speed $< 9.4\,\text{km}\,\text{h}^{-1}$, duration $> 45\,\text{h}$ (Houston, Dallas) Speed $< 8.3\,\text{km}\,\text{h}^{-1}$, duration $> 25\,\text{h}$ (Austin, San Antonio) | Centroid moves slowly and long duration |
| 4 | Warm-frontal storm | $V_N > 2\,\text{km}\,\text{h}^{-1}$ or $V_N > -2\,\text{km}\,\text{h}^{-1}$, $V_E < 0\,\text{km}\,\text{h}^{-1}$ | Clear motion northward or due west |
| 5 | Cold-frontal storm | $V_N < -2\,\text{km}\,\text{h}^{-1}$ or $V_N < 2\,\text{km}\,\text{h}^{-1}$, $V_E > 0\,\text{km}\,\text{h}^{-1}$ | Clear motion southward or due east |

Each storm is assigned to the first category that meets its criteria. $A_R$ and $A_{HR}$ denote the largest rainy ($\geq 20\,\text{dBZ}$) and heavy rain area ($\geq 40\,\text{dBZ}$) during each storm event; $V_N$ and $V_E$ represent the northerly and easterly components of the speed; duration denotes the time taken for each storm to traverse the research domain, determined by the duration of each storm event, the size of the research domain and the trajectory of the storms. An example of storm classification for 940 storms in Austin is shown in Supplementary Fig. 7.

**Extended Data Table 3 | Percentages of ≥40 dBZ (and ≥20 dBZ) grid cells contributed by each storm type**

| | Dallas | Austin | San Antonio | Houston |
|---|---|---|---|---|
| Single-cell storm | 7.5% (4.8%) | 10.0% (7.0%) | 9.9% (6.7%) | 13.5% (10.2%) |
| Isolated storm | 8.2% (5.8%) | 13.5% (11.9%) | 12.0% (10.6%) | 12.3% (9.2%) |
| Tropical system | 4.6% (9.5%) | 6.2% (11.7%) | 7.6% (11.2%) | 12.3% (19.1%) |
| Warm-frontal storm | 16.5% (19.6%) | 16.4% (18.8%) | 22.7% (27.2%) | 22.0% (25.7%) |
| Cold-frontal storm | 63.2% (60.2%) | 53.9% (50.6%) | 47.8% (44.4%) | 40.0% (35.8%) |

**Extended Data Table 4 | Average peak high-reflectivity area and storm duration across storm types**

| | Single-cell storm | Isolated storm | Tropical system | Warm-frontal storm | Cold-frontal storm |
|---|---|---|---|---|---|
| | High-reflectivity area at peak hour in km$^2$ (percent of city footprint) | | | | |
| Dallas | 47.4 (1.1%) | 401.4 (9.4%) | 6,138.4 (143.2%) | 4,444.8 (103.7%) | 6,409.0 (149.5%) |
| Austin | 51.8 (6.3%) | 417.8 (50.7%) | 2,198.9 (267.1%) | 1,807.3 (219.5%) | 3,013.9 (366.1%) |
| San Antonio | 47.6 (4.0%) | 409.9 (34.1%) | 3,114.7 (259.5%) | 2,350.8 (195.8%) | 3,422.3 (285.1%) |
| Houston | 45.5 (1.0%) | 332.5 (7.4%) | 9,435.5 (210.4%) | 4,604.2 (102.7%) | 6,180.4 (137.8%) |
| | Rainfall duration (h) | | | | |
| Dallas | 2.0 | 6.3 | 65.0 | 19.1 | 19.9 |
| Austin | 2.2 | 7.1 | 35.3 | 14.5 | 12.6 |
| San Antonio | 2.1 | 7.6 | 41.2 | 15.0 | 13.7 |
| Houston | 2.2 | 6.4 | 73.5 | 18.6 | 19.4 |