## [Peer Review file · Nature]

Divergent Urban Storm Response to Convective, Frontal, and Tropical Systems

Corresponding Author: Dr Xinxin Sui

Version 0:

Reviewer comments:

Referee #1

(Remarks to the Author)

Overall Comments:

The manuscript presents an event-based analysis to differentiate five storm types and investigate their associated urban effects using three-dimensional gridded radar reflectivity data. It aims to explore the divergent urban influences on distinct storm types through mechanism-specific discussions, concluding that urbanization enhances local-scale single-cell and isolated storms, weakens cold front storms, intensifies warm front storms, and has negligible impacts on tropical cyclones. The work contributes to advancing the understanding of urbanization's effects on extreme weather and contains certain innovative elements. The generalizability of the conclusions is influenced by several methodological considerations, which are outlined below.

Specific Comments:

1. Originality and Contributions

The manuscript exhibits originality in two key aspects:

It adopts a refined classification of storm types (single-cell, isolated, cold front, warm front, and tropical cyclones) within an event-based framework, focusing on the inherent characteristics of weather systems. This fills a research gap in understanding the impacts of urbanization on storms driven by specific weather systems, complementing existing studies that mainly focus on rainfall intensity, duration, or climatic zones.

The application of three-dimensional gridded radar reflectivity data is praiseworthy, as it allows for more accurate capture of the vertical structures and spatial distributions of storms compared to conventional satellite data, gauge observations, or model simulations.

2. Data Quality and Methodology

1) Data quality: Your manuscript lacks adequate discussion on the quality control of radar data. Given that the quality of radar data may vary spatially (e.g., interference in tropical cyclones), insufficient documentation of quality assurance procedures may undermine the reliability of storm type observations. It is recommended that you clarify the data processing steps, including how noise, attenuation, or terrain effects were mitigated.

2) Storm identification and classification:

(1) A flowchart illustrating the storm identification process would improve methodological transparency, helping readers evaluate reproducibility.

(2) Using uniform classification thresholds across cities with different sizes and climates may be inappropriate. Storm characteristics (e.g., intensity, spatial scale) inherently differ across regions, and a "one-size-fits-all" threshold may introduce biases into the results.

(3) Restricting the study area to a specific climate zone limits the generalizability of the conclusions, especially considering the known variations in storm type distributions across different climatic regions (e.g., tropical cyclones are concentrated in

low latitudes).

3. Statistical Methods

The lack of statistical significance tests (e.g., in Figs. 2–4) hinders the rigorous evaluation of urban-rural differences across the four cities. Incorporating appropriate tests (e.g., t-tests, ANOVA) to quantify the confidence in observed patterns is essential to enhance the robustness of the manuscript.

4. Mechanistic Explanations

Your manuscript claims that “urbanization affects different storm types through different mechanisms” but relies heavily on literature citations rather than original observational or modeling analyses to support these claims, which weakens the depth of mechanistic understanding. Existing research emphasizes that urbanization impacts precipitation through thermal (heat island), dynamic (airflow disturbances), and chemical (aerosol) processes. You should link your observational results to these core processes (e.g., how urban heat islands modulate convection in single-cell storms versus cold front systems) to consolidate the mechanistic interpretations.

5. Discussion and Contextualization

The discussion lacks sufficient integration with existing literature. For example: Existing studies (e.g., Xiong et al., 2025) note unique precipitation patterns in tropical monsoon regions, but your manuscript fails to explore potential linkages between its findings on tropical cyclones (negligible urban impact) and tropical monsoon dynamics (e.g., superimposed effects). Addressing such connections would enhance the contextual relevance of the study.

6. References

You are encouraged to incorporate the suggested references (Lalonde et al., 2023; Chang et al., 2025; Xiong et al., 2025; Shepherd, 2005) to strengthen contextualization, particularly regarding methodological diversity in urban precipitation research and cross-climatic zone comparisons.

Lalonde, M., Oudin, L., & Bastin, S. (2023). Urban effects on precipitation: Do the diversity of research strategies and urban characteristics preclude general conclusions? *Urban Climate*, 51, 101605. <https://doi.org/10.1016/j.uclim.2023.101605>

Chang, C.-C., Zhou, S., Cao, R., Yang, C., & Guo, H. (2025). Inconsistent urbanization effects on rainfall events across 19 urban agglomeration spanning humid to arid climates. *Journal of Hydrology*, 133969. <https://doi.org/10.1016/j.jhydrol.2025.133969>

Xiong, J., Yang, Y., Yang, L., et al. (2025). Asymmetric shifts in precipitation due to urbanization across global cities. *Nature Communications*, 16, 5802. <https://doi.org/10.1038/s41467-025-61053-0>

Shepherd, J. M. (2005). A review of current investigations of urban-induced rainfall and recommendations for the future. *Earth Interactions*, 9(12), 1–27. <https://doi.org/10.1175/EI156.1>

Earth Interactions, 9(12), 1–27. <https://doi.org/10.1175/EI156.1>

7. Technical Corrections

Line 412: “Table S1” should be corrected to “Table 1”.

8. Conclusion and Recommendation

The manuscript addresses an important topic with novel storm classification and data application. However, revisions are required to address the limitations in data documentation, methodological rigor, mechanistic depth, and contextual integration.

Referee #2

(Remarks to the Author)

In their paper, Sui et al. address one of the key unresolved questions in urban climatology: why some cities intensify rainfall while others suppress it. They propose a potential explanation by examining the urban impact on different rainfall types. This research question is both timely and of broad interest beyond the urban climate community, as the implications of rainfall intensification extend to flood risk reduction, transportation, energy systems, and other sectors. I consider the topic to fit well within the journal’s scope.

Overall

Overall, the paper is well written and logically structured. While some repetitions in the results and discussion sections could be reduced, the text is generally clear and easy to follow. The methodology is robust, although some statistical analyses are missing (see detailed comments below). The results are well presented and generally support the conclusions. Most of my comments concern textual refinements, with only a few relating to methodological aspects. I am confident these could be addressed promptly, and I hope to see the paper published thereafter.

I begin my review by outlining three major points. First, your storm archive contains only 9–26 tropical storms per city. This small sample size, in my view, limits the ability to draw firm conclusions about urban effects on tropical cyclones. I would consider removing TC from the results, but - if TC are retained in the classification for completeness, given their global relevance, I recommend being explicit about this limitation in the main text (currently hinted only in Table S1), exercising caution when interpreting urban–TC effects, and discussing this constraint in the discussion section.

Second, while I agree that the rainfall types identified in Texas are broadly representative of those found globally, cities in

other climatic regions (such as the tropics or semi-arid climates) may respond differently to e.g., cold fronts or single-cell storms. This issue is not fully addressed in the discussion. For example, could the cold-front mechanism illustrated in Fig. 5 operate in cities located in other climates? This warrants explicit discussion.

Third, as you are well aware, natural rainfall variability, especially in its spatial context, is high. Consequently, small differences between urban and rural domains (even changes that are of the order of 10%) may not be statistically significant. I recommend, where possible, applying statistical significance tests across the analysis. Some specific results where such tests are essential are indicated in my detailed comments below, but applying them broadly to other results, if possible, would surely strengthen the study.

Specific comments

Lines 40–41: This statement is not entirely accurate. Many studies focus on urbanization effects on convective rainfall or tropical cyclones in specific cities.

Lines 44–49: More precisely, the combination of the urban heat island and urban dry island increases atmospheric instability around the city (under weak wind conditions), which in turn enhances upward convection and fuels convective rainfall. There is a nice summary in a recent Urban Climate paper (<https://doi.org/10.1016/j.uclim.2025.102499>) that may be of interest.

Lines 89–90: Consider adding a supplementary figure showing representative radar examples for each of the five rainfall types.

Lines 137–138: Is the higher occurrence statistically significant, or could it fall within natural rainfall variability? Consider applying a statistical test to determine this.

Lines 144–150: The weekend–weekday analysis is interesting, but some differences (e.g., SC in Dallas, San Antonio, and Houston) may still fall within daily variability, even if reaching 12.7%. A statistical test should confirm which differences are significant.

Line 166: In this paragraph, you refer only to SC and IS, right? State this explicitly if so.

Line 173: “Dallas proper”?

Lines 281–283: Consider adding an illustration of these mechanisms as well in the supplementary material.

Lines 286–296: This paragraph repeats earlier points and I think it could be removed to keep the paper concise.

Radar data: I would encourage adding information on the reflectivity estimation errors and their influence on results. Also, can you provide supplementary maps showing areas affected by beam blockage or clutter, along with an explanation of how these were handled?

Urban domains: I think it would be valuable to examine the 1995 NLCD data (if available, if not, other data can be used) and check how well the domain extents match the 2019 NLCD definitions. Do any of the cities undergo substantial urban expansion between 1995 and 2019? This might influence the results presented.

Referee #3

(Remarks to the Author)

This manuscript investigates urban–storm interactions and develops a event-based analysis framework to explore whether different urban effects are associated with different storm types. By categorizing storms into five types (single-cell, isolated, tropical cyclones, warm fronts, and cold fronts), the authors identify distinct urban influences. The manuscript presents nice figures, and the number of storm cases is large. It is great to see the use of 3d radar data for studying the storms. Although the study domain is limited to four cities in Texas, it covers the major precipitation types observed in the region. Therefore, the study provides a useful framework that ranges from local to mesoscales and captures dominant storm types, offering valuable insights into urban–storm interactions. However, there are some major and minor concerns on the study listed below.

Major Comments:

1. Although five storm types are defined, the manuscript lacks detailed discussion of feasibility when defining the storms. The definitions of the storms are based on some criteria. For example, tropical cyclone is defined as: Speed < 9.4 km/h, Duration > 45 h (Houston, Dallas); or Speed < 8.3 km/h, Duration > 25 h (Austin, San Antonio). “Isolated storm” is defined as: AR < 7800 km² or AHR < 520 km². In other words, isolated storms are defined based on area, and tropical cyclone is defined by moving speed and duration. Warm front and cold front storm are defined by moving directions. This indicates that the different types of storms are classified by movement, duration and area, rather than their physical and meteorological settings commonly used in previous literature. For example, tropical cyclone should be defined by maximum sustained wind and warm cores. Frontal system should be identified by equivalent potential temperature gradient. The current definitions and tracking of the five types of storms should be justified.

2. While the authors argue that observational evidence is important, the authors might be aware of the challenges in deriving causal relationships using observations alone. Some experiments are useful to complement observational analyses. The authors wrote “Cold front storms weaken in urban areas (16–28% fewer intense rain areas), likely due to disruption by urban heat and surface roughness, while warm front storms have increased average reflectivity intensity” in the abstract. The statement would be stronger with well-designed experiments using regional climate models. The manuscript can also be strengthened by adding statistical significance test for all the figures in which comparisons are examined. Observational analysis is limited by mixing the influences of many factors that could affect the comparisons, relative to modeling experiments with controlled variables. Significance test is especially important in observational analysis to exclude potential effects of other factors. Some of the results could be artifacts without verification and statistical significance test. For example, the results shown in Figure 3 are hard to tell the significant differences made by urban effects in cities like Dallas, Austin and San Antonio. The differences might not necessarily be caused by urban effects given the fact that the differences can be affected by topography, seasons, wind, etc.

3. Interpretation of Fig. 3c and 3d: The results in Fig. 3c and 3d are particularly interesting: at low levels and during daytime, urban areas show a lower frequency of high reflectivity occurrence compared to rural areas. However, the authors do not provide an explanation or even mention this important finding. This result appears to be highly meaningful and should not be overlooked.

One possible mechanism is that aerosols and the urban heat island effect delay the onset of precipitation, shifting strong reflectivity to higher altitudes.

During daytime, enhanced surface heating over cities may strengthen boundary-layer dryness and turbulence, suppressing rapid droplet growth at low levels. In contrast, rural surfaces with stronger evapotranspiration (due to vegetation) provide more near-surface moisture, facilitating earlier precipitation onset. Therefore, the authors should explain why high reflectivity occurrence (40 dBZ above 1 km) in urban areas is consistently lower than in rural counterparts (Fig. 3c). Does this indeed suggest a delayed precipitation onset in urban environments? Similarly, for Fig. 3d, why is daytime high reflectivity occurrence lower in urban areas than in rural ones? A discussion of these mechanisms would strengthen the paper.

4. Inconsistencies in Lines 183–201 (interpretation of Fig. 3c and Fig. S4): The description in Lines 183–201 contains internal inconsistencies.

Fig. 3c focuses on small-scale single-cell and isolated storms, while Fig. S4 presents histograms of high reflectivity height distributions across all five storm types. However, the text combines these figures, which introduces contradictions. For example, the statement “While most high reflectivity grids are concentrated within 1–2 km in cities, over rural areas, they are 2–3 km” is not supported by Fig. 3c, which shows both urban and rural peaks near 2–3 km without a clear shift. Similarly, the statement “The average height decreases by 162 m (Houston) to 493 m (Austin)” contradicts earlier claims of “urban values being 180–260 m (or 0–400 m) higher than rural”. In fact, Fig. 3c shows urban curves shifted slightly higher at mid-levels, inconsistent with a decrease in mean height. Finally, attributing these results to “urban enhancement effect on tropical cyclones” is misleading, since Fig. 3c is based on single-cell and isolated convection, not tropical cyclones. The authors should carefully revise this section for internal consistency and ensure that the figure interpretations align with the storm types being discussed.

5. This study focuses on May–September for studying storms, including cold/warm fronts. It seems that the results on front storms are highlighted in the manuscript. However, the cold and warm fronts also occur commonly in the other months (e.g., winter). The front storms are different from tropical cyclones that occur mostly in the warm season. The front storms in the other months should be included, unless the authors want to exclude the results on fronts.

6. The merged three-dimensional GridRad dataset is useful to study 3d structure of the storms. Reflectivity is not a direct measurement of precipitation. Rather, it is a measurement of estimated hydrometeors in the atmosphere. 20/40 dBZ should be better justified to be used here when classifying storms. It seems that reflectivity data could be enhanced by using high-resolution precipitation data (e.g., Stage IV), from the impact perspective. Meanwhile, the atmospheric vertical profiles in reanalysis data should be analyzed to support the schematic in Figure 5.

Minor Comments:

1. The statement “However, for tropical storms, we find a consistent decrease in high reflectivity grid box height across four Texas cities” seems incorrect. Based on the figures, this consistent decrease appears in cold fronts, not tropical storms.

2. In Fig. S6c, the city boundary lines are difficult to distinguish from the color shading. The authors should consider using a colorbar with stronger contrast to improve visual clarity.

3. The manuscript needs careful proofreading. Below are some examples.

Line 14: “show” should be “showing”

Line 56: each what?

Line 59: short-duration

Line 68: making

Referee #4

(Remarks to the Author)

I co-reviewed this manuscript with one of the reviewers who provided the listed reports.

Version 1:

Reviewer comments:

Referee #1

(Remarks to the Author)

The authors have made careful revision and detailed response to the comments raised in the first review. I am pleased to see that most of the issues have been thoroughly addressed and significant improvements have been made to the manuscript. The explanations are clear and convincing, and the changes have enhanced the quality and readability of the paper.

However, I would still like to suggest a few minor adjustments to further refine the manuscript.

This research uses different speed and duration thresholds for large-scale tropical cyclones in different cities. It would be beneficial to provide an explanation on the applicability in other cities.

Additionally, the references are advised to be checked again for consistency in formatting.

Overall, I believe that with these final touches, the manuscript will be suited for publication in this journal.

Referee #2

(Remarks to the Author)

Thank you for addressing my comments from the first round of revisions and for updating the manuscript accordingly. While I am satisfied with your previous responses, after reading the revised version, I have a few additional minor comments and suggestions that I would like the authors to consider. These are all minor issues that I expect can be amended quickly. As none of them are methodological and they will not affect the results presented, I am happy to leave the verification of the revisions to the editor, and do not need to see the manuscript again before publication.

1. Line 71: Please briefly justify why the analysis focuses on the warm season only.

2. When reporting p-values, I recommend using a consistent notation and simply stating $p < 0.05$ for all cases where the p-value is significant.

3. Figure 2: Would it be useful to add schematic information on the main storm movement direction, or to indicate schematically the upwind and downwind directions for the cities shown?

4. Line 178: Please clarify why the threshold "above 1 km" is chosen.

5. Figures 3a and 3b: Here as well, it might be helpful to schematically distinguish between the upwind and downwind areas.

Referee #3

(Remarks to the Author)

The authors have addressed some of my previous comments. After re-reviewing the revised manuscript and the authors' responses to all the reviewers' comments, I have some remaining concerns.

1. As also pointed out by Referee#1, the authors mainly rely on previous literature to support the claims on mechanistical understanding and use merely 3D radar to discuss vertical distribution of hydrometeors. I want to emphasize that hydrometers might not necessarily be observed precipitation. NASA IMERG precipitation data might be able to complement the reflectivity data used in this study. Referee#1 also mentioned the three mechanisms underpinning urban effects in terms of thermal, dynamic and aerosol. This echoes my previous comments on adding physical experiments to verify the physical mechanisms. This additional step could add more confidence and provide a better context for future studies. Therefore, it might be helpful to add additional analyses of physical processes and regional model experiments for the storms, especially cold fronts, and this would dramatically strengthen the significance of this study. It might not be feasible to add experiments for all the storm cases, but some experiments for cold fronts when approaching the selected urban areas would be very insightful. Meanwhile, the physical diagnosis of some storms (e.g., cold fronts) using reanalysis data would also be desirable.

2. In Table S2 "Criteria for classifying storm type for the four cities in Texas", the authors use Speed and Duration to define tropical cyclone events. This might not be necessary because the National Hurricane Center has issued tropical cyclone data for the study period. The authors could also use tropical cyclone data from the National Hurricane Center to verify tropical cyclone events identified from their own definitions. Similarly, there are existing front data sets identified from reanalysis data, this could be very useful resources to check the robustness of the various types of storm systems especially cold fronts, which contribute to the most important findings of this study. Defining storms based on reflectivity could be enhanced by exploring synoptic conditions.

3. Figures S7-10 illustrate composite comparisons of cold front storms around four cities from one hour earlier to one hour later. While the differences in Dallas are remarkable, the results for other cities are slightly noisy, especially for the

enhancement of precipitation one hour earlier for cold fronts. The authors might need to reconcile the results. Can the authors provide more supporting evidence for Figure 4?

4. Please add an ensemble-sensitivity check in the Supplement to show that the main findings are not driven by a single subjective choice of thresholds. For example, it is necessary to rerun the storm classification using several plausible threshold sets by modestly perturbing only the most influential thresholds (e.g., the 20/40 dBZ reflectivity thresholds and the AR/AHR criteria), and demonstrate that the key conclusions for SC/IS/cold-front storms (direction and approximate magnitude of the urban–rural differences, and the main significance results) remain stable. In addition, since the paired city-year sample used for significance testing (N = 92) may not be fully independent due to spatial correlation across cities, please explicitly note this in the manuscript.

Referee #4

(Remarks to the Author)

I co-reviewed this manuscript with one of the reviewers who provided the listed reports.

Version 2:

Reviewer comments:

Referee #3

(Remarks to the Author)

The authors have made the efforts to improve the quality of this study. I have some remaining comments.

1. Line 511-517: The classification of storms in this study exhibits large uncertainties. I appreciate that the authors have checked the data in National Hurricane Center and found the discrepancy between “tropical cyclones” in previous version and the so-called “tropical systems” in this version. Only 14 hurricanes/tropical storms out of 26 events were “correctly” identified using the method. There might also be large uncertainties in cold or warm fronts. It seems that the authors did not expect such a significant discrepancy. That should motivate us to identify the storms of interest meteorologically and physically.
2. The reflectivity should be specified when discussed, because the reflectivity data used here is in 3D. Did the authors use maximum values or average values of reflectivity at the vertical level? I couldn't find the description in the methodology of the manuscript. For example, in Tables S2 and S3, the authors defined the kinds of storms based on AR and AHR. What does 40 dBZ represent here, maximum and average reflectivity at that spatial grid? Clarification would be very useful here.
3. Lines 82-84: “selecting those with high reflectivity areas (40 dBZ..)”. It might be helpful to specify what reflectivity is discussed. At which level? Maximum value? Average value? This would avoid confusion when reading the other parts of the manuscript.
4. Line 97: Please specify what reflectivity is used here. Same is at Line 126.
5. Lines 326-327: “We found more frequent local-scale single cell storms and isolated storms...” Where did you find more frequent storms described here and in what context?
6. Lines 266-267: It should be “The number of high reflectivity grid boxes increased as cold fronts approach the domains and decreased as they move away.”
7. Line 279: “before the city” might be “before the storm reaches the city”? Please clarify.

Referee #4

(Remarks to the Author)

I co-reviewed this manuscript with one of the reviewers who provided the listed reports.

Referee #1 (Remarks to the Author):

Overall Comments:

The manuscript presents an event-based analysis to differentiate five storm types and investigate their associated urban effects using three-dimensional gridded radar reflectivity data. It aims to explore the divergent urban influences on distinct storm types through mechanism-specific discussions, concluding that urbanization enhances local-scale single-cell and isolated storms, weakens cold front storms, intensifies warm front storms, and has negligible impacts on tropical cyclones. The work contributes to advancing the understanding of urbanization's effects on extreme weather and contains certain innovative elements. The generalizability of the conclusions is influenced by several methodological considerations, which are outlined below.

Specific Comments:

1. Originality and Contributions

The manuscript exhibits originality in two key aspects:

It adopts a refined classification of storm types (single-cell, isolated, cold front, warm front, and tropical cyclones) within an event-based framework, focusing on the inherent characteristics of weather systems. This fills a research gap in understanding the impacts of urbanization on storms driven by specific weather systems, complementing existing studies that mainly focus on rainfall intensity, duration, or climatic zones.

The application of three-dimensional gridded radar reflectivity data is praiseworthy, as it allows for more accurate capture of the vertical structures and spatial distributions of storms compared to conventional satellite data, gauge observations, or model simulations.

Thank you for recognizing the originality and novelty of our work.

2. Data Quality and Methodology

1) Data quality: Your manuscript lacks adequate discussion on the quality control of radar data. Given that the quality of radar data may vary spatially (e.g., interference in tropical cyclones), insufficient documentation of quality assurance procedures may undermine the reliability of storm type observations. It is recommended that you clarify the data processing steps, including how noise, attenuation, or terrain effects were mitigated.

We appreciate the reviewer for bringing up this point. We have added more description about the GridRad data processing steps and quality control methods in the Methods–Radar reflectivity data section. The revised text is copied below for your reference:

“... This study directly analyzes reflectivity rather than rain rate (R), thereby avoiding uncertainties in the Z - R relationship, which is one of the main sources of uncertainty in radar-based precipitation retrievals... GridRad data has been widely used to analyze mesoscale convective systems^{43,44}, hail⁴⁵, and tropopause-overshooting convection⁴⁶. It merges reflectivity data from 125 National Weather Service NEXRAD WSR-88D Level 2 weather radars⁴⁷ onto a common three-dimensional grid through a four-step algorithmic procedure: 1) reading raw data, 2) identifying grid volumes, 3) computing space–time weights, and 4) applying weighted binning⁴¹. In addition to the space–time weighting scheme that reduces noise from low-quality or distant data, GridRad applies several quality control procedures, such as filtering low-confidence echoes and removing ground clutter, to minimize errors from non-meteorological sources. Details of the quality control process and validation against other radar datasets are described in the algorithm description document⁴¹. To further reduce residual noise caused by remaining artifacts, this research applies two reflectivity thresholds to identify storms: 20 dBZ to differentiate rain from drizzle, very light rain, or non-precipitating echoes, and 40 dBZ

(corresponding to a rain rate of about 10 mm per hour) to identify heavy rainfall^{38,48}. The identified storms with areas smaller than 100 km² are excluded to further minimize uncertainties associated with small-scale artifacts.”

2) Storm identification and classification:

(1) A flowchart illustrating the storm identification process would improve methodological transparency, helping readers evaluate reproducibility.

We have added a flowchart as Fig. S12 in the latest version of manuscript.

Fig. S12: Flowchart illustrating storm identification and post-processing procedures.

(2) Using uniform classification thresholds across cities with different sizes and climates may be inappropriate. Storm characteristics (e.g., intensity, spatial scale) inherently differ across regions, and a “one-size-fits-all” threshold may introduce biases into the results.

As the reviewer notes in their next comment, four cities share broadly similar subtropical climates. More importantly, the interaction of storms with cities depends on the property of cities rather than what storms are “typical” for each city. This research uses different thresholds for large-scale tropical cyclones and uniform thresholds for convective and frontal storms (as detailed below). This setup is based on our evaluation of storm performance across the observation windows, which showed that local-scale convective and frontal storms exhibit similar behavior in the domains. However, due to the much larger spatial scale of tropical cyclones, the size of the observation domain affected storm performance differently across different size of cities. Therefore, it is more appropriate to use different thresholds for large and medium-sized cities, which were calibrated separately. The final classification algorithm is copied below for your reference.

Table S2: Criteria for classifying storm type for the four cities in Texas.

Classification order	Storm type	Definition	Character
1	Single cell storm	AR < 1,300 km ² , Duration ≤ 5 h	Small rain area, short duration

2	Isolated storm	$AR < 7,800 \text{ km}^2$ or $AHR < 520 \text{ km}^2$	Relatively small rain and heavy rain area
3	Tropical cyclone	$Speed < 9.4 \text{ km/h}$, $Duration > 45 \text{ h}$ (Houston, Dallas) $Speed < 8.3 \text{ km/h}$, $Duration > 25 \text{ h}$ (Austin, San Antonio)	Centroid moves slowly and long duration
4	Warm front storm	$VN > 2 \text{ km/h}$ or $VN > -2 \text{ km/h}$, $VE < 0 \text{ km/h}$	Clear motion northward or due west
5	Cold front storm	$VN < -2 \text{ km/h}$ or $VN < 2 \text{ km/h}$, $VE > 0 \text{ km/h}$	Clear motion southward or due east

(3) Restricting the study area to a specific climate zone limits the generalizability of the conclusions, especially considering the known variations in storm type distributions across different climatic regions (e.g., tropical cyclones are concentrated in low latitudes).

Our study area features a combination of midlatitude and tropical influences, allowing us to analyze storms ranging from local convective to frontal and tropical systems, which are commonly observed worldwide. Second, Texas, with its wide climatic and geographical variation, includes climate ranging from humid subtropical to semi-arid, covering both coastal and inland cities, as well as flat and hilly terrains. The consistent urban-storm effects identified across these contrasting landscapes suggest that our findings are robust and broadly applicable beyond Texas. However, we indeed missed several important storm types, such as orographic rainfall, monsoon storms⁴⁰, or mesoscale convective systems. The limitation has been stated in the revised manuscript.

“In addition, this study focuses only on warm-season and does not include storm types including orographic rainfall, monsoon storms⁴⁰, or mesoscale convective systems. Future studies are encouraged to incorporate diverse weather systems and regional contexts, as these factors governs dominant storm types and mechanisms, and consequently, the ways cities module regional rainfall patterns...”

40. Xiong, J., Yang, Y., Yang, L. and Yang, D., 2025. Asymmetric shifts in precipitation due to urbanization across global cities. *Nature Communications*, 16(1), p.5802.”

3. Statistical Methods

The lack of statistical significance tests (e.g., in Figs. 2–4) hinders the rigorous evaluation of urban-rural differences across the four cities. Incorporating appropriate tests (e.g., t-tests, ANOVA) to quantify the confidence in observed patterns is essential to enhance the robustness of the manuscript.

Thank you for this suggestion. In the revised manuscript, we have added significance tests to strength the urban–rural comparisons. For local scale convective storms, where we compare the occurrence of storms (Fig. 2 and Table S5) and high reflectivity grids (Fig. 3), we apply the Paired Wilcoxon signed-rank test. To ensure sufficient data pairs, we use yearly data from 1995–2017 across the four cities, resulting in $N = 92$ (4 cities \times 23 years). For frontal storms, where we compare the individual storm intensity (Fig. 4), each frontal storms can be considered as independent, and therefore we are able to apply the non-paired Wilcoxon signed-rank test.

“Fig. 2: Deviation in the number of five types of warm-season storms in four Texas cities compared to the average storm number in their peripheral rural control areas (1995-

2017). Paired Wilcoxon signed-rank tests on yearly storm counts ($n = 92$) confirm that single cell storms (SC) and isolated storms (IS) occur significantly more frequently in urban areas than their rural counterparts, while no significant differences are detected for other storm types (Table S5). A similar plot demonstrating the deviation in the number of high reflectivity grid boxes (40 dBZ) contributed by five types of storms is shown in Fig. S2.

Fig. 3: Urban enhancement of local-scale single cell storms and isolated storms. (a, b) Spatial anomalies in high reflectivity occurrence (40 dBZ, above 1 km) in urban areas compared to their rural counterparts. The red polygon marks the urban boundary, and the anomalies within this boundary are indicated in the figures. Paired Wilcoxon signed-rank tests confirm a significant increase in yearly high reflectivity occurrence in urban areas for single cell storms ($p = 0.0026$, $r = 0.31$, $n = 92$) and a marginal increase for isolated storms ($p = 0.056$, $r = 0.20$, $n = 92$). The spatial comparisons for the other three storm types are shown in Fig. S5. (c) Vertical and (d) diurnal probability distribution (local time) of high reflectivity grids for urban and rural storms. Urban anomalies above 5 km or before 9 am (shaded area) are quantified in the legend. For Houston, as the southern rural domain lies over the ocean, where rainfall patterns differ from those on land, we exclude the data from the southern rural area when calculating the rural comparison.”

Fig. 4: Composite comparisons between urban and rural cold front storm intensities. (a) Vertical profiles for average cold front storms passing through cities (solid line) and four surrounding rural areas (blue-shaded zone). Significantly fewer high reflectivity grid boxes are found in the lower atmosphere (2–6 km) of urban domains compared to rural counterparts, with anomaly and Wilcoxon signed-rank test results indicated. The n_u and n_r values indicate the number of cold front storms passing urban and rural domains, respectively. (b) Dynamical

comparison of urban and rural cold fronts approaching and leaving urban and rural domains. When cold fronts approach cities, a larger fraction of cold front storms enter the observation window, leading to an increase in 40 dBZ grid boxes (above the zero line) and vice versa as they leave. We found that cities may attenuate cold front intensity at its nearest approach and slightly enhance intensity ahead of arrival. Non-parametric Wilcoxon signed-rank tests confirm significant decreases in high reflectivity grid boxes as cold fronts reach most cities (except Dallas, $p = 0.057$), while the increases ahead of cities are not significant.

Table S5: Paired Wilcoxon signed-rank test results for urban–rural differences in storm numbers. The null hypothesis (H_0) is that the yearly storm number in urban areas is not greater than the average of their four rural counterparts. 23 years (1995–2017) observations across four cities yield 92 paired data for sufficient statistical power ($n = 92$).”

	W	p value	r value
Single cell storm	2673	0.0059	0.29
Isolated storm	2691	0.0011	0.34
Tropical cyclone	1141.5	0.41	0.086
Warm front	1907	0.36	0.095
Cold front	1813.5	0.60	0.055

4. Mechanistic Explanations

Your manuscript claims that “urbanization affects different storm types through different mechanisms” but relies heavily on literature citations rather than original observational or modeling analyses to support these claims, which weakens the depth of mechanistic understanding. Existing research emphasizes that urbanization impacts precipitation through thermal (heat island), dynamic (airflow disturbances), and chemical (aerosol) processes. You should link your observational results to these core processes (e.g., how urban heat islands modulate convection in single-cell storms versus cold front systems) to consolidate the mechanistic interpretations.

Thank you for raising this important point. We acknowledge that a purely observation-based approach limits our ability to directly verify physical mechanisms. To strengthen the mechanistic interpretations, we have incorporated insights from three complementary perspectives: 1) previous numerical simulation on urban rainfall effects (10,18,36); 2) storm dynamical studies (37), and 3) observations studies (39,40). Synthesizing these studies with our findings, we are able to enhance the mechanistic understanding of urban storm impacts in the revised manuscript.

This study reveals three types of urban effects related to different mechanisms: 1) the thermal effect (urban heat islands) on enhanced convective rainfall found in previous studies is consistent with the behavior of convective storms studied here, 2) dynamic and surface roughness effects that contribute to cold front attenuation and tropical cyclone modification are supported by some simulations but require further studies, and 3) the mechanisms behind urban influences on warm front storms remain largely unexplored, highlighting the need for additional numerical and observation analysis. All of these provide valuable directions for future numerical studies. The revised manuscript is copied below for your reference.

“Results show that all four cities exhibit a higher occurrence of isolated storms than their rural counterparts...The mechanism behind this has been documented in numerous simulation studies. Previous studies have shown that urban heat islands enhance atmospheric convection

and promote the development of local-scale convection-driven storms in densely populated areas^{9,10}...

The increase in high-level reflectivity leads to a rise in average storm height...These results provide observational verification of the stronger and deeper convection over urban areas reported in previous numerical studies^{10,33}...

When the storms are at their closest approach to the city center, all four cities show varying degrees of reduction in the probability of high reflectivity grids, with some areas reaching a 10% reduction. One hour earlier, we found an approximately 9% increase in the probability of high reflectivity grid boxes to the northwest of Austin and San Antonio... Gaffen and Bornstein (1988)³⁵ investigated a moving cold front passing through New York City and reported that urban roughness can disrupt the lower part of a cold front. Hu et al. (2025) found a similar attenuation-intensification pattern when cold fronts approach and pass through the Greater Bay Area³⁶. Their numerical simulations confirm the variations in frontal intensity corresponding to changes in the meridional gradient of equivalent potential temperature at 925 hPa...

This study reveals that urbanization's effects are not uniform but vary with storm mechanisms and scales. We found more frequent local-scale single-cell storms and isolated storms, particularly at night. These results are consistent with previous numerical studies that demonstrate UHI-driven thermally enhanced convection^{8,9,10}. For frontal storms, urbanization can influence intensity rather than frequency: cold fronts are significantly weakened over cities as increased surface roughness and urban heat disrupt their dynamic structure and thermal gradients, while warm fronts show moderate but nonsignificant intensification. Numerical simulations for New York City and the Greater Bay Areas attribute these changes in cold frontal intensity to both dynamic and thermal factors^{35,36}, whereas few studies have documented any urban-induced changes in warm frontal storms. This contrast could be related to different structures and mechanisms of the two frontal storms: cold frontal storms are typically narrow, deep convective systems, while warm frontal storms are dominated by broad stratiform clouds, making the latter less sensitive to land surface perturbations³⁷.

5. Discussion and Contextualization

The discussion lacks sufficient integration with existing literature. For example: Existing studies (e.g., Xiong et al., 2025) note unique precipitation patterns in tropical monsoon regions, but your manuscript fails to explore potential linkages between its findings on tropical cyclones (negligible urban impact) and tropical monsoon dynamics (e.g., superimposed effects). Addressing such connections would enhance the contextual relevance of the study.

We appreciate the reviewer for sharing this interesting study. We have added further discussion linking our findings with up-to-date literatures and the research limitation. The revised discussion, which better contextualizes our results within the broader scientific context, is copied below for your reference.

"Tropical cyclones show no changes in frequency or intensity but shift high reflectivity grids at lower altitudes over urban areas, possibly due to surface roughness. This finding, based on three-dimensional radar reflectivity data, contrasts with previous simulation results that reported urban intensification^{11,12}. The discrepancy may arise from differences between observational analysis and numerical simulation, or from uncertainties associated with limited sample size. Texas typically experiences about one tropical cyclone per year, which constrains our ability to detect significant urban effects within the 23-year dataset... Capturing storm types ranging from convective to frontal and tropical systems, this study helps reconcile the distinct urban effects reported in previous studies^{13,14}. Chang et al. (2025)³⁹ differentiated short-term and long-term rainfall events and found enhanced short-duration rainfall but suppressed long-duration rainfall

over cities. Our results support and extend these findings by providing storm dynamics evidence linking short-term rainfall to local-scale convective storms and long-term rainfall to synoptic frontal systems that persist longer. This study focuses on four Texas cities that encompass diverse climatic and geographic settings. Dallas and Houston are relatively flat, while Austin and San Antonio have more pronounced topographic variation. Houston is coastal, while the other three are inland. Despite these contrasts, we find consistent urban storm effects across four cities, indicating robust city–storm mechanisms. Nonetheless, Texas’s climate variation remains limited and cannot represent all global conditions. Although we did not explicitly address broader regional climate forcings, our findings, together with previous research²⁶, suggest that urbanization effects do not vanish under stronger regional drivers; rather, urban and regional impacts often coexist. An example from this study is Houston. Proximity to the warm Gulf of Mexico limits the strength of cold frontal systems, making the urban weakening effect on cold frontal storms less apparent than other inland cities. In addition, this study focuses only on warm-season and does not include storm types including orographic rainfall, monsoon storms⁴⁰, or mesoscale convective systems. Future studies are encouraged to incorporate diverse weather systems and regional contexts, as these factors governs dominant storm types and mechanisms, and consequently, the ways cities modulate regional rainfall patterns.”

6. References

You are encouraged to incorporate the suggested references (Lalonde et al., 2023; Chang et al., 2025; Xiong et al., 2025; Shepherd, 2005) to strengthen contextualization, particularly regarding methodological diversity in urban precipitation research and cross-climatic zone comparisons.

Lalonde, M., Oudin, L., & Bastin, S. (2023). Urban effects on precipitation: Do the diversity of research strategies and urban characteristics preclude general conclusions? *Urban Climate*, 51, 101605. <https://doi.org/10.1016/j.uclim.2023.101605>

Chang, C.-C., Zhou, S., Cao, R., Yang, C., & Guo, H. (2025). Inconsistent urbanization effects on rainfall events across 19 urban agglomeration spanning humid to arid climates. *Journal of Hydrology*, 133969. <https://doi.org/10.1016/j.jhydrol.2025.133969>

Xiong, J., Yang, Y., Yang, L., et al. (2025). Asymmetric shifts in precipitation due to urbanization across global cities. *Nature Communications*, 16, 5802. <https://doi.org/10.1038/s41467-025-61053-0>

Shepherd, J. M. (2005). A review of current investigations of urban-induced rainfall and recommendations for the future. *Earth Interactions*, 9(12), 1–27. <https://doi.org/10.1175/EI156.1>

We thank the reviewer for suggesting these references and have added them as references [6, 14, 39, and 40] in the Introduction and Discussion.

7. Technical Corrections

Line 412: “Table S1” should be corrected to “Table 1”.

This is corrected in the latest manuscript.

8. Conclusion and Recommendation

The manuscript addresses an important topic with novel storm classification and data application. However, revisions are required to address the limitations in data documentation, methodological rigor, mechanistic depth, and contextual integration.

We sincerely appreciate the reviewer’s constructive suggestions, which have helped us improve the quality of this research.

Referee #2 (Remarks to the Author):

In their paper, Sui et al. address one of the key unresolved questions in urban climatology: why some cities intensify rainfall while others suppress it. They propose a potential explanation by examining the urban impact on different rainfall types. This research question is both timely and of broad interest beyond the urban climate community, as the implications of rainfall intensification extend to flood risk reduction, transportation, energy systems, and other sectors. I consider the topic to fit well within the journal's scope.

Overall, the paper is well written and logically structured. While some repetitions in the results and discussion sections could be reduced, the text is generally clear and easy to follow. The methodology is robust, although some statistical analyses are missing (see detailed comments below). The results are well presented and generally support the conclusions. Most of my comments concern textual refinements, with only a few relating to methodological aspects. I am confident these could be addressed promptly, and I hope to see the paper published thereafter.

We appreciate your comments and suggestions in helping us improve our manuscript. I begin my review by outlining three major points. First, your storm archive contains only 9–26 tropical storms per city. This small sample size, in my view, limits the ability to draw firm conclusions about urban effects on tropical cyclones. I would consider removing TC from the results, but - if TC are retained in the classification for completeness, given their global relevance, I recommend being explicit about this limitation in the main text (currently hinted only in Table S1), exercising caution when interpreting urban–TC effects, and discussing this constraint in the discussion section.

Thank you for pointing this out. We agree that this point is important to mention earlier in the manuscript. In the latest version, we have highlighted the limited number of tropical cyclones in the main text and further emphasized it as part of the limitations in the discussion section as suggested. The added paragraph is copied below for your reference.

Main text: *“we find a consistent, though not statistically significant, decrease in the average height of tropical cyclones across four Texas cities... Although not statistically significant due to the limited number of tropical cyclones observed in Texas over the 23-year study period, this downward shift is noteworthy and needs further study of tropical cyclone structure and intensity using larger event samples¹¹”.*

Discussion: *“Tropical cyclones show no changes in frequency or intensity but shift high reflectivity grids at lower altitudes over urban areas... Texas typically experiences about one tropical cyclone per year, which constrains our ability to detect significant urban effects within the 23-year dataset”.*

Second, while I agree that the rainfall types identified in Texas are broadly representative of those found globally, cities in other climatic regions (such as the tropics or semi-arid climates) may respond differently to e.g., cold fronts or single-cell storms. This issue is not fully addressed in the discussion. For example, could the cold-front mechanism illustrated in Fig. 5 operate in cities located in other climates? This warrants explicit discussion.

We appreciate the reviewer for pointing this out and understand this concern. Our previous studies on urban rainfall across global and U.S. cities showed that climatic conditions (such as tropics or semi-arid climates) could influence the magnitude of urban rainfall anomalies (larger anomalies under warmer and wetter climates) but do not reverse or offset the urban effects. We noted that this is based on a long-term climatological analysis rather than individual storm dynamics.

For the cold-front mechanism illustrated in Fig. 5, this figure indicates the dual influences of urban heat islands and surface roughness on cold front storms. Roughness is a structural characteristic of urbanization, mainly related to building height and density. Therefore, the roughness should persist across different climates. As we mentioned in the manuscript, a numerical study on cold fronts over New York, which has a different climate from Texas, also reported reduce storm intensity. However, we know the second factor, urban heat islands, can weaken or even reverse in semi-arid regions. Under this circumstance, we could expect a reduced urban effect especially for the slightly enhanced intensity in Fig. 5a. In summary, this mechanism is expected to operate across different climates as long as the key factors, urban roughness and heat islands, are present.

However, we acknowledge that this is not a global study, and the mechanisms have not been verified in other climatic regions. To address this, we have added a limitation in the Discussion section and elaborated on the interplay between urbanization and regional conditions:

“Nonetheless, Texas’s climate variation remains limited and cannot represent all global conditions. Although we did not explicitly address broader regional climate forcings, our findings, together with previous research²⁶, suggest that urbanization effects do not vanish under stronger regional drivers; rather, urban and regional impacts often coexist. An example from this study is Houston. Proximity to the warm Gulf of Mexico limits the strength of cold frontal systems, making the urban weakening effect on cold frontal storms less apparent than other inland cities”.

Third, as you are well aware, natural rainfall variability, especially in its spatial context, is high. Consequently, small differences between urban and rural domains (even changes that are of the order of 10%) may not be statistically significant. I recommend, where possible, applying statistical significance tests across the analysis. Some specific results where such tests are essential are indicated in my detailed comments below, but applying them broadly to other results, if possible, would surely strengthen the study.

Thank you for this suggestion. In the revised manuscript, we have added significance tests to strength the urban–rural comparisons. For local scale convective storms, where we compare the occurrence of storms (Fig. 2 and Table S5) and high reflectivity gris (Fig. 3), we apply the Paired Wilcoxon signed-rank test. To ensure sufficient data pairs, we use yearly data from 1995–2017 across the four cities, resulting in $N = 92$ (4 cities \times 23 years). For frontal storms, where we compare the individual storm intensity (Fig. 4), each frontal storms can be considered as independent, and therefore we are able to apply the non-paired Wilcoxon signed-rank test.

“Fig. 2: Deviation in the number of five types of warm-season storms in four Texas cities compared to the average storm number in their peripheral rural control areas (1995-2017). Paired Wilcoxon signed-rank tests on yearly storm counts ($n = 92$) confirm that single cell storms (SC) and isolated storms (IS) occur significantly more frequently in urban areas than their rural counterparts, while no significant differences are detected for other storm types (Table S5). A similar plot demonstrating the deviation in the number of high reflectivity grid boxes (40 dBZ) contributed by five types of storms is shown in Fig. S2.”

“Fig. 3: Urban enhancement of local-scale single cell storms and isolated storms. (a, b) Spatial anomalies in high reflectivity occurrence (40 dBZ, above 1 km) in urban areas compared to their rural counterparts. The red polygon marks the urban boundary, and the anomalies within this boundary are indicated in the figures. Paired Wilcoxon signed-rank tests confirm a significant increase in yearly high reflectivity occurrence in urban areas for single cell storms ($p = 0.0026$, $r = 0.31$, $n = 92$) and a marginal increase for isolated storms ($p = 0.056$, $r = 0.20$, $n = 92$). The spatial comparisons for the other three storm types are shown in Fig. S5. (c) Vertical

and (d) diurnal probability distribution (local time) of high reflectivity grids for urban and rural storms. Urban anomalies above 5 km or before 9 am (shaded area) are quantified in the legend. For Houston, as the southern rural domain lies over the ocean, where rainfall patterns differ from those on land, we exclude the data from the southern rural area when calculating the rural comparison.”

“Fig. 4: Composite comparisons between urban and rural cold front storm intensities. (a) Vertical profiles for average cold front storms passing through cities (solid line) and four surrounding rural areas (blue-shaded zone). Significantly fewer high reflectivity grid boxes are found in the lower atmosphere (2–6 km) of urban domains compared to rural counterparts, with anomaly and Wilcoxon signed-rank test results indicated. The *n_u* and *n_r* values indicate the number of cold front storms passing urban and rural domains, respectively. (b) Dynamical comparison of urban and rural cold fronts approaching and leaving urban and rural domains. When cold fronts approach cities, a larger fraction of cold front storms enter the observation window, leading to an increase in 40 dBZ grid boxes (above the zero line) and vice versa as they leave. We found that cities may attenuate cold front intensity at its nearest approach and slightly enhance intensity ahead of arrival. Non-parametric Wilcoxon signed-rank tests confirm significant decreases in high reflectivity grid boxes as cold fronts reach most cities (except Dallas, *p* = 0.057), while the increases ahead of cities are not significant.”

“Table S5: Paired Wilcoxon signed-rank test results for urban–rural differences in storm numbers. The null hypothesis (*H₀*) is that the yearly storm number in urban areas is not greater than the average of their four rural counterparts. 23 years (1995–2017) observations across four cities yield 92 paired data for sufficient statistical power (*n* = 92).”

	W	p value	r value
Single cell storm	2673	0.0059	0.29
Isolated storm	2691	0.0011	0.34
Tropical cyclone	1141.5	0.41	0.086
Warm front	1907	0.36	0.095
Cold front	1813.5	0.60	0.055

Specific comments

Lines 40–41: This statement is not entirely accurate. Many studies focus on urbanization effects on convective rainfall or tropical cyclones in specific cities.

We have revised this sentence. The revised version is as follows.

“Many previous studies have examined storm-specific urban impacts through numerical simulations for case studies, particularly on convective rainfall^{8,9,10} and tropical cyclones^{11,12}. However, variations across storm types and mechanisms remain underexplored in large sample observational datasets.”

Lines 44–49: More precisely, the combination of the urban heat island and urban dry island increases atmospheric instability around the city (under weak wind conditions), which in turn enhances upward convection and fuels convective rainfall. There is a nice summary in a recent Urban Climate paper (<https://doi.org/10.1016/j.uclim.2025.102499>) that may be of interest.

We have revised this corresponding sentence and incorporated the suggested paper as ref. 10. The revised version is provided below.

“The mechanism behind this is that urban heat islands lead to increased atmospheric instability over cities¹⁰, which enhances upward convection and fuels convective rainfall⁸. Under weak wind conditions, the rainfall enhancement tends to be in the cities, while strong winds can shift the rainfall hotspots to the downwind areas¹⁹.”

Lines 89–90: Consider adding a supplementary figure showing representative radar examples for each of the five rainfall types.

We have added two supplementary figures as shown below. We excluded the smallest single-cell type as its small spatiotemporal scale makes its less visible as other types.

Fig S12. Hourly snapshots of selected isolated, warm front, cold front storm cases passing Austin. Animations are available for download from Zenodo (10.5281/zenodo.15933280).

Fig S13. Hourly snapshots of a selected tropical cyclone case passing Austin. Animations are available for download from Zenodo (10.5281/zenodo.15933280).

Lines 137–138: Is the higher occurrence statistically significant, or could it fall within natural rainfall variability? Consider applying a statistical test to determine this.

We have added the significant test results in Table S5 and the corresponding analysis in the main text: “We used the non-parametric Paired Wilcoxon signed-rank test to evaluate the significance of differences in urban and rural storm numbers (Table S5). The results confirm that

urban areas have significantly more frequent local-scale single cell storms ($p = 0.0059$, $r = 0.29$, $n = 92$) and isolated storms ($p = 0.0011$, $r = 0.34$, $n = 92$).

Lines 144–150: The weekend–weekday analysis is interesting, but some differences (e.g., SC in Dallas, San Antonio, and Houston) may still fall within daily variability, even if reaching 12.7%. A statistical test should confirm which differences are significant.

We have added the significant test for the weekend–weekday comparison as shown below.

Fig S4. Histogram of the height distribution of high reflectivity grid boxes (40 dBZ) in urban and rural domains for five storm types. The rural histogram represents the average result across four rural domains. The majority of these histograms shows more high reflectivity grids at 1 km in cities than rural domain, which may be related to uncertainties in low-level radar measurements caused by urban ground clutter. Comparing the average height of high reflectivity grid boxes (vertical lines) with the lowest 1 km excluded, single cell storms are consistently and significantly higher in urban areas compared to rural areas across four cities (Paired Wilcoxon signed-rank test: $p < 0.001$, $r = 0.57$, $n = 92$), whereas tropical cyclones are consistently but not significantly lower ($p = 0.18$, $r = 0.21$, $n = 40$). The south domain data over the ocean are excluded for Houston.

Line 166: In this paragraph, you refer only to SC and IS, right? State this explicitly if so.

Yes, we have revised this part to explicitly mention single cell storms and isolated storms.

“Fig. 3a and 3b show the spatial anomalies above 1 km for single cell storms and isolated storms.”

Line 173: “Dallas proper”?

We have modified this part into “although positive anomalies are shown in the eastern half of the metropolitan area where Dallas is located”

Lines 281–283: Consider adding an illustration of these mechanisms as well in the supplementary material.

This part refers to the warm front intensification. Since this change is not statistically significant, unlike the findings for local convective storms and cold fronts, we prefer to treat this cautiously without adding a schematic illustration to avoid potential misleading knowledge propagation.

Lines 286–296: This paragraph repeats earlier points and I think it could be removed to keep the paper concise.

We agree on this suggestion. The repeated sentences are deleted in the latest version of manuscript.

Radar data: I would encourage adding information on the reflectivity estimation errors and their influence on results. Also, can you provide supplementary maps showing areas affected by beam blockage or clutter, along with an explanation of how these were handled?

We appreciate the reviewer’s suggestion. We have revised the radar data section to included additional information about data processing and quality control. Since this study uses a gridded radar product (GridRad) that merges reflectivity data from multiple nearby radars rather than a single radar source, much of the noise and spatial bias have already been mitigated. Low-confidence echoes and ground clutter are filtered out through the GridRad quality control procedures. The revised section is copied below for your reference.

“It merges reflectivity data from 125 National Weather Service NEXRAD WSR-88D Level 2 weather radars⁴⁷ onto a common three-dimensional grid through a four-step algorithmic

procedure: 1) reading raw data, 2) identifying grid volumes, 3) computing space–time weights, and 4) applying weighted binning⁴¹. In addition to the space–time weighting scheme that reduces noise from low-quality or distant data, GridRad applies several quality control procedures, such as filtering low-confidence echoes and removing ground clutter, to minimize errors from non-meteorological sources. Details of the quality control process and validation against other radar datasets are described in the algorithm description document⁴¹. To further reduce residual noise caused by remaining artifacts, this research applies two reflectivity thresholds to identify storms: 20 dBZ to differentiate rain from drizzle, very light rain, or non-precipitating echoes, and 40 dBZ (corresponding to a rain rate of about 10 mm per hour) to identify heavy rainfall^{38,48}. The identified storms with areas smaller than 100 km² are excluded to further minimize uncertainties associated with small-scale artifacts.”

Urban domains: I think it would be valuable to examine the 1995 NLCD data (if available, if not, other data can be used) and check how well the domain extents match the 2019 NLCD definitions. Do any of the cities undergo substantial urban expansion between 1995 and 2019? This might influence the results presented.

We appreciate the reviewer for bringing up this point. We have checked the 1995 land cover type data to evaluate how well the urban domain extents align with the 2019 NLCD definitions. This figure below shows a different extent of urban expansion across these cities. However, this expansion does not lead to major domain changes. This research use 2019 domain allowing us to capture a larger number of storms for the urban–rural comparison.

Referee #3 (Remarks to the Author):

This manuscript investigates urban–storm interactions and develops a event-based analysis framework to explore whether different urban effects are associated with different storm types. By categorizing storms into five types (single-cell, isolated, tropical cyclones, warm fronts, and cold fronts), the authors identify distinct urban influences. The manuscript presents nice figures, and the number of storm cases is large. It is great to see the use of 3d radar data for studying the storms. Although the study domain is limited to four cities in Texas, it covers the major precipitation types observed in the region. Therefore, the study provides a useful framework that ranges from local to mesoscales and captures dominant storm types, offering valuable insights into urban–storm interactions. However, there are some major and minor concerns on the study listed below.

Major Comments:

1. Although five storm types are defined, the manuscript lacks detailed discussion of feasibility when defining the storms. The definitions of the storms are based on some criteria. For example, tropical cyclone is defined as: Speed < 9.4 km/h, Duration > 45 h (Houston, Dallas); or Speed < 8.3 km/h, Duration > 25 h (Austin, San Antonio). “Isolated storm” is defined as: AR < 7800 km² or AHR < 520 km². In other words, isolated storms are defined based on area, and tropical cyclone is defined by moving speed and duration. Warm front and cold front storm are defined by moving directions. This indicates that the different types of storms are classified by movement, duration and area, rather than their physical and meteorological settings commonly used in previous literature. For example, tropical cyclone should be defined by maximum sustained wind and warm cores. Frontal system should be identified by equivalent potential temperature gradient. The current definitions and tracking of the five types of storms should be justified.

Thank you for this constructive comment. We acknowledge that the current storm classification is primarily based on storm performance rather than purely mechanism-based definitions. We designed this approach as it balances between classification accuracy and computational feasibility, allowing us to objectively process a large number of storm events while maintaining meaningful distinctions among storm types.

Since storms of different meteorological origins exhibit consistent and recognizable performance features, storm behavior has been used as a proxy for classification in previous research. Lorenz et al., (2018) has used wetted area ratio and heavy-rain coverage (Table shown below) to classify storms near Berlin, German. We extended this framework by incorporating additional dynamic properties, such as storm velocity and duration, to improve classification robustness.

We have also integrated physical and synoptic information during the calibration stage. Specifically, we examine NOAA national forecast charts, wind fields, and temperature changes before and after storms passages for a subset of storms near classification thresholds. While such verification was not feasible for all storms due to the large dataset size, these checks help us estimate reasonable thresholds for the storm calibration. We have modified our storm classification for greater clarity and discussed the limitation behind within the discussion section.

Table 1**Definition of Precipitation Types** According to Two Indices WAR and 10AR

Precipitation types		Definition [-]		
Cold fronts ^a		$0.10 < \text{WAR} < 0.50$	and	$0 < 10\text{AR} < 0.25$
Warm fronts ^a	Advection ^b	$0.25 < \text{WAR} < 1$	and	$0 < 10\text{AR} < 0.05$
Showers ^a		$0 < \text{WAR} < 0.15$	and	$0 < 10\text{AR} < 0.05$
Mesoscale convective systems ^a	Convection ^b	$0.10 < \text{WAR} < 0.50$	and	$0.05 < 10\text{AR} < 0.50$
Convective cells ^a	Convection ^b	$0 < \text{WAR} < 0.15$	and	$0.10 < 10\text{AR} < 0.70$

Note. Wetted area index (WAR) is the proportion of rainfall exceeding 1 mm/hr; area ratio of rain intensity above 10 mm/hr (10AR) is the rainfall proportion of WAR exceeding 10 mm/hr.

^aEhret (2003). ^bJatho et al. (2010).

Ref: Lorenz, J.M., Kronenberg, R., Bernhofer, C. and Niyogi, D., 2019. Urban rainfall modification: Observational climatology over Berlin, Germany. *Journal of Geophysical Research: Atmospheres*, 124(2), pp.731-746.

“By subjectively observing the radar animations of storms, we roughly estimate the initial maximum and minimum thresholds to determine an uncertainty band. For those storms within the uncertainty band, we further examine the synoptic situations and label storm types by referencing the National Oceanic and Atmospheric Administration (NOAA) weather archive of the national forecast charts⁵⁰, wind conditions, and temperature changes before and after storms. With a group of storms with labeled storm types, we are able to calibrate the exact thresholds for these factors and refine the storm classification method (Table S2). In this way, we develop an objective storm classification method (Fig. S11), which is subsequently applied to all storm events.”

“Limitations of this study could arise from the uncertainty in the storm classification method. The objective classification algorithm developed in this study builds upon the earlier subjective classification approach by Lorenz et al. (2019)³⁸. In addition to the rainfall and heavy rainfall areas considered in that work, our method incorporates more dynamic characteristics, including storm movement and duration. This classification framework relies primarily on storm behavior and dynamic properties. Given the large sample size, this study does not examine synoptic conditions for every case, such as temperature gradients or sustained wind. Instead, we only investigate synoptic conditions for a limited number of storms near the classification boundary to calibrate the objective criteria for storm performance and dynamics. This compromise between classification accuracy and computational feasibility enables us to transit from subjective classification to a fully automated, objective framework applicable to a large number of storm events. Strictly speaking, this algorithm is based more directly on storm characteristics than pre-defined storm type categories, though it aligns well with common recognized storm types. We expect this approach to be suited for studying storm-urban interactions, as it relies on intrinsic storm properties that govern how storms respond to urban environments.”

2. While the authors argue that observational evidence is important, the authors might be aware of the challenges in deriving causal relationships using observations alone. Some experiments are useful to complement observational analyses. The authors wrote “Cold front storms weaken in urban areas (16–28% fewer intense rain areas), likely due to disruption by urban heat and surface roughness, while warm front storms have increased average reflectivity intensity” in the abstract. The statement would be stronger with well-designed experiments using regional climate models. The manuscript can also be strengthened by adding statistical significance test for all the figures in which comparisons are examined. Observational analysis is limited by mixing the influences of many factors that could affect the comparisons, relative to modeling

experiments with controlled variables. Significance test is especially important in observational analysis to exclude potential effects of other factors. Some of the results could be artifacts without verification and statistical significance test. For example, the results shown in Figure 3 are hard to tell the significant differences made by urban effects in cities like Dallas, Austin and San Antonio. The differences might not necessarily be caused by urban effects given the fact that the differences can be affected by topography, seasons, wind, etc.

We appreciated reviewer for raising these two points and fully agree with the comments. In the revised manuscript, we have (i) incorporated previous numerical simulation studies to strengthen the mechanism discussion and (ii) added statistical tests to support the robustness of our findings. The revised sections of the manuscript are copied below for your reference.

(i) “Results show that all four cities exhibit a higher occurrence of isolated storms than their rural counterparts...The mechanism behind this has been documented in numerous simulation studies. Previous studies have shown that urban heat islands enhance atmospheric convection and promote the development of local-scale convection-driven storms in densely populated areas^{9,10}...”

The increase in high-level reflectivity leads to a rise in average storm height...These results provide observational verification of the stronger and deeper convection over urban areas reported in previous numerical studies^{10,33}...

When the storms are at their closest approach to the city center, all four cities show varying degrees of reduction in the probability of high reflectivity grids, with some areas reaching a 10% reduction. One hour earlier, we found an approximately 9% increase in the probability of high reflectivity grid boxes to the northwest of Austin and San Antonio...Gaffen and Bornstein (1988)³⁵ investigated a moving cold front passing through New York City and reported that urban roughness can disrupt the lower part of a cold front. Hu et al. (2025) found a similar attenuation-intensification pattern when cold fronts approach and pass through the Greater Bay Area³⁶. Their numerical simulations confirm the variations in frontal intensity corresponding to changes in the meridional gradient of equivalent potential temperature at 925 hPa...

This study reveals that urbanization’s effects are not uniform but vary with storm mechanisms and scales. We found more frequent local-scale single-cell storms and isolated storms, particularly at night. These results are consistent with previous numerical studies that demonstrate UHI-driven thermally enhanced convection^{8,9,10}. For frontal storms, urbanization can influence intensity rather than frequency: cold fronts are significantly weakened over cities as increased surface roughness and urban heat disrupt their dynamic structure and thermal gradients, while warm fronts show moderate but nonsignificant intensification. Numerical simulations for New York City and the Greater Bay Areas attribute these changes in cold frontal intensity to both dynamic and thermal factors^{35,36}, whereas few studies have documented any urban-induced changes in warm frontal storms. This contrast could be related to different structures and mechanisms of the two frontal storms: cold frontal storms are typically narrow, deep convective systems, while warm frontal storms are dominated by broad stratiform clouds, making the latter less sensitive to land surface perturbations³⁷.”

(ii) “Fig. 2: Deviation in the number of five types of warm-season storms in four Texas cities compared to the average storm number in their peripheral rural control areas (1995-2017). Paired Wilcoxon signed-rank tests on yearly storm counts (n = 92) confirm that single cell storms (SC) and isolated storms (IS) occur significantly more frequently in urban areas than their rural counterparts, while no significant differences are detected for other storm

types (Table S5). A similar plot demonstrating the deviation in the number of high reflectivity grid boxes (40 dBZ) contributed by five types of storms is shown in Fig. S2.”

“Fig. 3: Urban enhancement of local-scale single cell storms and isolated storms. (a, b) Spatial anomalies in high reflectivity occurrence (40 dBZ, above 1 km) in urban areas compared to their rural counterparts. The red polygon marks the urban boundary, and the anomalies within this boundary are indicated in the figures. Paired Wilcoxon signed-rank tests confirm a significant increase in yearly high reflectivity occurrence in urban areas for single cell storms ($p = 0.0026$, $r = 0.31$, $n = 92$) and a marginal increase for isolated storms ($p = 0.056$, $r = 0.20$, $n = 92$). The spatial comparisons for the other three storm types are shown in Fig. S5. (c) Vertical and (d) diurnal probability distribution (local time) of high reflectivity grids for urban and rural storms. Urban anomalies above 5 km or before 9 am (shaded area) are quantified in the legend. For Houston, as the southern rural domain lies over the ocean, where rainfall patterns differ from those on land, we exclude the data from the southern rural area when calculating the rural comparison.”

“Fig. 4: Composite comparisons between urban and rural cold front storm intensities. (a) Vertical profiles for average cold front storms passing through cities (solid line) and four surrounding rural areas (blue-shaded zone). Significantly fewer high reflectivity grid boxes are found in the lower atmosphere (2–6 km) of urban domains compared to rural counterparts, with anomaly and Wilcoxon signed-rank test results indicated. The n_u and n_r values indicate the number of cold front storms passing urban and rural domains, respectively. (b) Dynamical comparison of urban and rural cold fronts approaching and leaving urban and rural domains. When cold fronts approach cities, a larger fraction of cold front storms enter the observation window, leading to an increase in 40 dBZ grid boxes (above the zero line) and vice versa as

they leave. We found that cities may attenuate cold front intensity at its nearest approach and slightly enhance intensity ahead of arrival. Non-parametric Wilcoxon signed-rank tests confirm significant decreases in high reflectivity grid boxes as cold fronts reach most cities (except Dallas, $p = 0.057$), while the increases ahead of cities are not significant.”

“Table S5: Paired Wilcoxon signed-rank test results for urban–rural differences in storm numbers. The null hypothesis (H_0) is that the yearly storm number in urban areas is not greater than the average of their four rural counterparts. 23 years (1995–2017) observations across four cities yield 92 paired data for sufficient statistical power ($n = 92$).”

	W	p value	r value
Single cell storm	2673	0.0059	0.29
Isolated storm	2691	0.0011	0.34
Tropical cyclone	1141.5	0.41	0.086
Warm front	1907	0.36	0.095
Cold front	1813.5	0.60	0.055

3. Interpretation of Fig. 3c and 3d: The results in Fig. 3c and 3d are particularly interesting: at low levels and during daytime, urban areas show a lower frequency of high reflectivity occurrence compared to rural areas. However, the authors do not provide an explanation or even mention this important finding. This result appears to be highly meaningful and should not be overlooked.

One possible mechanism is that aerosols and the urban heat island effect delay the onset of precipitation, shifting strong reflectivity to higher altitudes.

During daytime, enhanced surface heating over cities may strengthen boundary-layer dryness and turbulence, suppressing rapid droplet growth at low levels. In contrast, rural surfaces with stronger evapotranspiration (due to vegetation) provide more near-surface moisture, facilitating earlier precipitation onset. Therefore, the authors should explain why high reflectivity occurrence (40 dBZ above 1 km) in urban areas is consistently lower than in rural counterparts (Fig. 3c). Does this indeed suggest a delayed precipitation onset in urban environments? Similarly, for Fig. 3d, why is daytime high reflectivity occurrence lower in urban areas than in rural ones? A discussion of these mechanisms would strengthen the paper.

We appreciate the reviewer for sharing these possible mechanisms. Fig. 3c and 3d show possibility density functions rather than histogram, so the urban and rural distributions cannot be directly compared by magnitude. These two figures are build based on Fig 3a and 3b that show increased high reflectivity grids in four cities. Fig. 3c and 3d further illustrate the largest increase among the vertical (Fig 3c) and diurnal (Fig 3d) changes. To avoid potential misinterpretation, we revised the manuscript flow and added a histogram as Fig. S6 to provide a clearer comparison. This additional figure confirms that the urban enhancement of high reflectivity could occur across a day.

Fig. S6: Histogram of the diurnal distribution of high reflectivity grid boxes (40 dBZ) in urban and rural domains for five storm types. Time is displayed in local time.

4. Inconsistencies in Lines 183–201 (interpretation of Fig. 3c and Fig. S4): The description in Lines 183–201 contains internal inconsistencies.

Fig. 3c focuses on small-scale single-cell and isolated storms, while Fig. S4 presents histograms of high reflectivity height distributions across all five storm types. However, the text combines these figures, which introduces contradictions.

For example, the statement “While most high reflectivity grids are concentrated within 1–2 km in cities, over rural areas, they are 2–3 km” is not supported by Fig. 3c, which shows both urban and rural peaks near 2–3 km without a clear shift.

Similarly, the statement “The average height decreases by 162 m (Houston) to 493 m (Austin)” contradicts earlier claims of “urban values being 180–260 m (or 0–400 m) higher than rural”. In

fact, Fig. 3c shows urban curves shifted slightly higher at mid-levels, inconsistent with a decrease in mean height. Finally, attributing these results to “urban enhancement effect on tropical cyclones” is misleading, since Fig. 3c is based on single-cell and isolated convection, not tropical cyclones.

The authors should carefully revise this section for internal consistency and ensure that the figure interpretations align with the storm types being discussed.

We are grateful to the reviewer for identifying this potential misleading statement. This part, “*While most high reflectivity grids are concentrated within 1–2 km in cities, over rural areas, they are 2–3 km*”, was intended to refer to the tropical cyclone in Figure S4 rather than the isolated and single cell storms shown in Figure 3. Accordingly, we have revised the relevant section to make a clear distinction between isolated storms and tropical cyclones. The revised text appears below.

“Based on previous results showing more frequent local-scale storms and more high reflectivity grids in urban areas, we further examine the vertical probability distribution to identify the main layers of enhancement (Fig. 3c). In general, 80% to 90% of high reflectivity grids are below 5 km, with peaks around 2–3 km for two types of local-scale storms. However, when comparing urban and rural domains, the majority of the increase appears aloft: above 5 km, all cities show increases in high reflectivity grid boxes ranging from 21% to 59% for inland cities and 134%–159% for Houston. The increase in high-level reflectivity leads to a rise in average storm height: single cell storms are elevated by 186–423 m in cities, while isolated storms show changes of -64–525 m with the lowest 1 km of atmosphere excluded (Fig. S4)... For other storm types, we did not observe significant or consistent changes in high reflectivity height across four cities, except for tropical cyclones: we find a consistent, though not statistically significant, decrease in the average height of tropical cyclones across four Texas cities. Over rural areas, most high reflectivity grid boxes are concentrated at 2–3 km, whereas they shift downward to 1–2 km when passing over cities (Fig. S4c). The average height of the high reflectivity grid boxes decreases by 13 m in Houston to 304 m in Austin, even when the lowest 1 km is excluded.”

5. This study focuses on May–September for studying storms, including cold/warm fronts. It seems that the results on front storms are highlighted in the manuscript. However, the cold and warm fronts also occur commonly in the other months (e.g., winter). The front storms are different from tropical cyclones that occur mostly in the warm season. The front storms in the other months should be included, unless the authors want to exclude the results on fronts.

The current research setup is based on our previous findings that urban rainfall influences are more pronounced during the summer than in other seasons. To better capture these apparent urbanization effects, this study is designed to focus on the warm season, leaving the October–April out of our study period. We have revised the discussion section to explicitly incorporating this point as a potential limitation.

“In addition, this study focuses only on warm-season and does not include storm types including orographic rainfall, monsoon storms⁴⁰, or mesoscale convective systems. Future studies are encouraged to incorporate diverse weather systems and regional contexts, as these factors governs dominant storm types and mechanisms, and consequently, the ways cities modulate regional rainfall patterns”.

6. The merged three-dimensional GridRad dataset is useful to study 3d structure of the storms. Reflectivity is not a direct measurement of precipitation. Rather, it is a measurement of estimated hydrometeors in the atmosphere. 20/40 dBZ should be better justified to be used here when classifying storms. It seems that reflectivity data could be enhanced by using high-resolution precipitation data (e.g., Stage IV), from the impact perspective. Meanwhile, the

atmospheric vertical profiles in reanalysis data should be analyzed to support the schematic in Figure 5.

Our previous work has used IMERG and Stage IV precipitation datasets to investigate precipitation differences. Although these datasets provided quantitative estimates of rainfall amounts, they did not reveal storm mechanisms nor vertical structures. This motivated us to use radar data in the present study. Radar data provides a comprehensive, three-dimensional view of storms, and low-level radar reflectivity is highly correlated with precipitation intensity. While rain gauges directly measure precipitation, they capture only a tiny footprint of the overall storm, leading to major sampling deficiencies. We have also examined reanalysis data before, however, such data often fail to capture urban conditions such as heat island effects or rainfall modifications due to lack advanced urban parameterizations. This highlights the significance of observational analyses such as this study, which help to inform and improve future modeling efforts.

Minor Comments:

1. The statement “However, for tropical storms, we find a consistent decrease in high reflectivity grid box height across four Texas cities” seems incorrect. Based on the figures, this consistent decrease appears in cold fronts, not tropical storms.

This sentence should refer to Fig. S4. We have revised this figure to include a vertical line indicating the average height. With this, we could visualize the lower heights of tropical cyclones over cities better.

Fig. S4: Histogram of the height distribution of high reflectivity grid boxes (40 dBZ) in urban and rural domains for five storm types. The rural histogram represents the average result across four rural domains. The majority of these histograms show more high reflectivity grids at 1 km altitude in cities than rural domain, which may be related to uncertainties in low-level radar measurements caused by ground blockage and clutter. Comparing the average height of high reflectivity grid boxes (vertical lines) with the lowest 1 km excluded, single cell storms are consistently and significantly higher in urban areas compared to rural areas across four cities (Paired Wilcoxon signed-rank test: $p < 0.001$, $r = 0.57$, $n = 92$), whereas tropical cyclones are consistently but not significantly lower ($p = 0.18$, $r = 0.21$, $n = 40$). The south domain data over the ocean are excluded for Houston.

2. In Fig. S6c, the city boundary lines are difficult to distinguish from the color shading. The authors should consider using a colorbar with stronger contrast to improve visual clarity. We appreciate this suggestion. We have changed the color of city boundary lines from red to black.

Fig. S5: Spatial anomalies of high reflectivity grid boxes (40 dBZ) from (a) tropical cyclone, (b) warm front, and (c) cold front storms around Texas cities compared to their rural counterparts, similar to Fig. 3a and Fig. 3b for single cell storms and isolated storms.

3. The manuscript needs careful proofreading. Below are some examples.

Line 14: “show” should be “showing”

Line 56: each what?

Line 59: short-duration

Line 68: making

We have carefully proofread the manuscript and revised these issues in the latest version.

Referee #4 (Remarks to the Author):

I co-reviewed this manuscript with one of the reviewers who provided the listed reports.

We appreciate your comments and suggestions in improving the manuscript quality.

Referee #1 (Remarks to the Author):

The authors have made careful revision and detailed response to the comments raised in the first review. I am pleased to see that most of the issues have been thoroughly addressed and significant improvements have been made to the manuscript. The explanations are clear and convincing, and the changes have enhanced the quality and readability of the paper.

Thank you for your suggestions in last round of revisions, which have helped us improve this manuscript.

However, I would still like to suggest a few minor adjustments to further refine the manuscript.

This research uses different speed and duration thresholds for large-scale tropical cyclones in different cities. It would be beneficial to provide an explanation on the applicability in other cities.

Thanks for this suggestion. We have added a new section called “*Sensitivity and Uncertainty Analyses*” in the revised manuscript. In this section, we mention the applicability to other regions:

“This classification criteria and specific thresholds used in our study are calibrated based on storm characteristics in our observation windows. These thresholds should be recalibrated when applying this approach to other regions, particular for identifying large frontal and tropical systems. Such adaptations should account for regional differences in storm pathways, typical trajectory, spatiotemporal scale, and local climatological characteristics.”

Additionally, the references are advised to be checked again for consistency in formatting.

We have checked and revised the reference list.

Overall, I believe that with these final touches, the manuscript will be suited for publication in this journal.

Referee #2 (Remarks to the Author):

Thank you for addressing my comments from the first round of revisions and for updating the manuscript accordingly. While I am satisfied with your previous responses, after reading the revised version, I have a few additional minor comments and suggestions that I would like the authors to consider. These are all minor issues that I expect can be amended quickly. As none of them are methodological and they will not affect the results presented, I am happy to leave the verification of the revisions to the editor, and do not need to see the manuscript again before publication.

1. Line 71: Please briefly justify why the analysis focuses on the warm season only.

The reason is provided in lines 70–72, “*Given that previous studies have highlighted a significant urban effect on summer precipitation compared to other seasons²⁶, this study focuses on intense warm-season (May – September) storms.*”

2. When reporting p-values, I recommend using a consistent notation and simply stating $p < 0.05$ for all cases where the p-value is significant.

Thank you for this suggestion. We have checked the author guidelines (“*Where relevant, provide exact values for both significant and non-significant P values*”). We decide to consistently show the exact p values in the manuscript when space is available, which provide more information than a threshold or range.

3. Figure 2: Would it be useful to add schematic information on the main storm movement direction, or to indicate schematically the upwind and downwind directions for the cities shown?

Among five types of storms, the first two types of convective storms are characterized by short duration and highly variable trajectories, without a clear moving direction (see figure in response #5). We therefore did not include schematic storm directions for these cases. For cold frontal storms, the movement is shown in Fig S7 to S10. For warm front storms and tropical systems, we did not include upwind and downwind trajectories as individual storms exhibit substantially different pathways, and no significant urban effects were identified for them.

4. Line 178: Please clarify why the threshold “above 1 km” is chosen.

The reason is provided in lines 173–174:” *Here, we exclude reflectivity data below 1 km altitude from the spatial comparison to reduce potential radar errors near ground.*”

5. Figures 3a and 3b: Here as well, it might be helpful to schematically distinguish between the upwind and downwind areas.

Fig. 3a and 3b refer to the two types of convective storms. As shown in Table S4, the average duration of single cell storms is only two hours, which is not sufficient for a robust upwind-downwind analysis. We did investigate the trajectories of isolated storms. However, their movement directions are highly variable, without a clearly preferred direction. We are providing an example showing 158 isolated storms in Austin and their average hourly displacements in the figure below.

Referee #3 (Remarks to the Author):

The authors have addressed some of my previous comments. After re-reviewing the revised manuscript and the authors’ responses to all the reviewers’ comments, I have some remaining concerns.

We sincerely appreciate your rigorous altitude and constructive suggestions, which have helped us improve the quality of this research.

1. As also pointed out by Referee#1, the authors mainly rely on previous literature to support the claims on mechanistical understanding and use merely 3D radar to discuss vertical distribution of hydrometeors. I want to emphasize that hydrometers might not necessarily be observed

precipitation. NASA IMERG precipitation data might be able to complement the reflectivity data used in this study. Referee#1 also mentioned the three mechanisms underpinning urban effects in terms of thermal, dynamic and aerosol. This echoes my previous comments on adding physical experiments to verify the physical mechanisms. This additional step could add more confidence and provide a better context for future studies. Therefore, it might be helpful to add additional analyses of physical processes and regional model experiments for the storms, especially cold fronts, and this would dramatically strengthen the significance of this study. It might not be feasible to add experiments for all the storm cases, but some experiments for cold fronts when approaching the selected urban areas would be very insightful. Meanwhile, the physical diagnosis of some storms (e.g., cold fronts) using reanalysis data would also be desirable.

In our earlier works (7,26), we have used NASA IMERG and Stage IV precipitation datasets to quantify the urban–rural precipitation differences, which motivated us to further investigate the storm structure using three-dimensional radar reflectivity in the current work.

We appreciate your suggestion on numerical simulations to investigate cold front effects. As noted in the manuscript, similar numerical experiments have been conducted in previous studies across regions (35,36). However, these efforts have not led to a broadly accepted theoretical understanding due to substantial model uncertainty and limited sample sizes. Recognizing these limitations, the primary objective of this study is to provide a robust observational perspective based on a large number of events and long-term observations, rather than introducing another case-specific simulation. We hope the reviewer could agree that this large-sample observational study represents an important and unique step toward guiding future modeling efforts and laying the groundwork for broader physical understanding on different urban–storm interactions.

2. In Table S2 “Criteria for classifying storm type for the four cities in Texas”, the authors use Speed and Duration to define tropical cyclone events. This might not be necessary because the National Hurricane Center has issued tropical cyclone data for the study period. The authors could also use tropical cyclone data from the National Hurricane Center to verify tropical cyclone events identified from their own definitions. Similarly, there are existing front data sets identified from reanalysis data, this could be very useful resources to check the robustness of the various types of storm systems especially cold fronts, which contribute to the most important findings of this study. Defining storms based on reflectivity could be enhanced by exploring synoptic conditions.

We appreciate this constructive suggestion. While this classification does not represent a perfect storm taxonomy, it provides an effective and efficient framework for distinguishing local convection, frontal storms, and tropical events for urban and rural comparisons. The consistent storm indicators (area, duration, and velocity) are applied across all storms to systematically differentiate major weather system types, rather than to optimize detection of any specific storm type. Although we avoid using reanalysis data due to their limited ability to resolve urban precipitation processes, we agree that National Hurricane Center records provide a reliable reference for tropical systems. Given the relatively small number of tropical cases, we are able to evaluate the 26 tropical systems identified in Houston using independent references, including National Hurricane Center records, Texas historical climate reports, and additional literatures. We found that, among 26 events, 14 were officially named hurricanes or tropical storms, and 8 were other weather systems influenced by tropical airflow. Based on this verification, we revised the manuscript to consistent use the broader terminology “*tropical systems*”, rather than strictly “*tropical cyclones*” throughout the manuscript. The added figure and analysis (lines 508–521) are provided below for your reference.

“We further evaluated 26 tropically influenced weather systems identified in Houston, using National Hurricane Center tropical cyclone reports, Texas Climate Report, and additional sources (Table S7). Among these 26 events, 14 were officially named hurricanes or tropical storms, while 8 were other weather systems influenced by tropical air flow, including tropical waves, tropical influenced convective storm, and mixed weather systems that perform similar to tropical storms within our observation window. This outcome is expected, as our classification emphasizes long event durations and large, heavy-rainfall footprints, which are typically associated with tropical moisture. Overall, the results indicate that our algorithms could reliably identifies storms influenced by tropical systems. This classification criteria and specific thresholds used in our study are calibrated based on storm characteristics in our observation windows. These thresholds should be recalibrated when applying this approach to other regions, particular for identifying large frontal and tropical systems. Such adaptations should account for regional differences in storm pathways, typical trajectory, spatiotemporal scale, and local climatological characteristics.”

Table S7. Verification of 26 tropical influenced weather system identified in Houston. The start dates and duration indicate the periods when each event entered the observation window, rather than the full lifecycle of the weather system. In some cases, long-lived hurricanes or tropical storms are separated into two events due to spatial or temporal discontinuities within the observational window.

	Start date	Weather event	Duration (h)	Reference
1	6/28/95	Tropically influenced convective system	53	
2	8/23/96	Hurricane Dolly	47	Tropical Cyclone Reports
3	8/28/96	Hurricane Dolly	87	Tropical Cyclone Reports
4	7/30/97	Isolated convective storms	53	
5	8/6/97	Stationary front	47	Texas Climate Report
6	9/21/97	Tropically influenced cold front storm	101	Texas Climate Report
7	9/8/98	Tropical Storm Frances	127	Tropical Cyclone Reports
8	9/14/98	Tropical Storm Frances	71	Tropical Cyclone Reports
9	6/6/01	Tropical Storm Allison	54	Tropical Cyclone Reports
10	6/8/01	Tropical Storm Allison	50	Tropical Cyclone Reports
11	8/28/01	Tropical Storm Dean	138	Tropical Cyclone Reports
12	9/22/01	Cold Front	48	Texas Climate Report
13	7/15/02	Tropical wave	50	National Weather Service
14	8/14/02	Tropically influenced convective system	51	Texas Climate Report
15	9/20/03	Tropically influenced MCS	60	
16	5/13/04	Tropically influenced MCS	46	Curtis et al., 2004
17	6/25/04	Subtropical jet influenced frontal storm	65	Texas Climate Report
18	7/3/07	Tropically enhanced upper-level trough	66	Texas Climate Report
19	9/10/09	Hurricane Ike	94	Fanelli et al., 2009
20	6/29/10	Hurricane Alex	91	Tropical Cyclone Reports
21	9/6/10	Tropical Storm Hermine	70	Tropical Cyclone Reports
22	9/19/13	Hurricane Ingrid	56	Tropical Cyclone Reports
23	6/16/15	Tropical Storm Bill	71	Tropical Cyclone Reports
24	8/13/16	Tropical easterly wave	96	Brown et al., 2020
25	6/21/17	Tropical Storm Cindy	66	Tropical Cyclone Reports
26	8/24/17	Hurricane Harvey	154	Tropical Cyclone Reports

3. Figures S7-10 illustrate composite comparisons of cold front storms around four cities from one hour earlier to one hour later. While the differences in Dallas are remarkable, the results for

other cities are slightly noisy, especially for the enhancement of precipitation one hour earlier for cold fronts. The authors might need to reconcile the results. Can the authors provide more supporting evidence for Figure 4?

Thank you for this comment. The enhancement of precipitation one hour earlier is not obvious, which is consistent with our expectations, as Fig 4b also shows the enhancement signal is not statistically significant. To reduce potential misunderstanding and improve consistency between Fig 4 and Fig S7–S10, we have applied a Gaussian smoothing filter to Fig S7–S10 to emphasize the smoothed differences and highlighted the statistically significant results in the caption.

Fig. S7: Composite comparisons of cold front storms around Austin. The spatial map shows the frequency of high reflectivity grid boxes upon arrival and one hour before and after arrival in urban or rural domains, with cold fronts moving from the northwest toward the southwest. The difference between urban and rural cold front composite is shown after applying a two-dimensional Gaussian smoothing filter. It indicates a significant reduction in reflectivity intensity over Austin upon arrival time, and a non-significant increase in high reflectivity grid boxes

northeast of Austin one hour before. The red polygon marks the Austin urban boundary within the observation window for reference.

Fig. S8: Composite comparisons of cold front storms around San Antonio, similar to Fig. S7. San Antonio shows a similar cold front modification pattern as Austin. Upon arrival, we found a significant reduction in reflectivity intensity over San Antonio and non-significant increase in high reflectivity grids northeast of the city one hour before.

Fig. S9: Composite comparisons of cold front storms around Dallas, similar to Fig. S7 and S8. With a much larger city footprint than Austin and San Antonio, cold front storms show attenuation one hour before reaching the Dallas city center.

Fig. S10: Composite comparisons of cold front storms around Houston, similar to Fig. S7, S8, and S9. Apparently due to the southern location of Houston compared to other large Texas cities and its proximity to the Gulf Coast in the southeast, cold fronts weaken greatly before reaching Houston without a distinct frontal shape in the figure.

4. Please add an ensemble-sensitivity check in the Supplement to show that the main findings are not driven by a single subjective choice of thresholds. For example, it is necessary to rerun the storm classification using several plausible threshold sets by modestly perturbing only the most influential thresholds (e.g., the 20/40 dBZ reflectivity thresholds and the AR/AHR criteria), and demonstrate that the key conclusions for SC/IS/cold-front storms (direction and approximate magnitude of the urban–rural differences, and the main significance results) remain stable. In addition, since the paired city-year sample used for significance testing ($N = 92$) may not be fully independent due to spatial correlation across cities, please explicitly note this in the manuscript.

We have added an additional section, “*Sensitivity and Uncertainty Analyses*”, in the revised manuscript to test the robustness of our finding to thresholds choices and to incorporate the

tropical cyclone verification in earlier response #2. The added section (lines 497–508) is copied below for your reference:

“Sensitivity and Uncertainty Analyses

...To assess the robustness of storm identification and the corresponding urban effects, we conducted sensitivity tests using higher radar reflectivity thresholds. Since the 20 dBZ value corresponds to very light precipitation, lower reflectivity levels are inconsequential to the main conclusions; therefore, we use elevated thresholds of 25 and 45 dBZ to repeat the analysis. Storm counts in Table S6 shows that local-scale single cell and isolated storms are more sensitive to threshold selection, with approximately 40% and 20% fewer storm events. In contrast, larger-scale frontal storms and tropical systems exhibit greater robustness to threshold changes. Despite the reduced sample size, the urban–rural storm number comparison remains significant for single cell and isolated storms (Table S6). However, because these local-scale storms may be driven by the same mesoscale or synoptic-scale systems and exhibit spatial correlation, this could reduce the effective degrees of freedom in the paired city–year sample, introducing uncertainty into the significant test. For cold front storms, the previous significant decrease in intensity over urban areas also remains robust (Fig S.16)... ”

Table S6. Sensitivity of urban–rural storm number comparisons to reflectivity thresholds. Similar to Table S1 and S5, it shows storm counts identified with elevated reflectivity thresholds (25 and 45 dBZ) and the significant test. The null hypothesis (H_0) is that the yearly storm number in urban areas is not greater than the average of their four rural counterparts.

	Dallas	Austin	San Antonio	Houston	W	p value	r value
Single cell storms	931 (63.0%)	268 (43.2%)	333 (49.9%)	2609 (74.9%)	2717	0.0068	0.28
Isolated storms	194 (13.1%)	114 (18.4%)	109 (16.3%)	433 (12.4%)	2665	0.0033	0.31
Tropical system	10 (0.7%)	16 (2.6%)	9 (1.3%)	26 (0.7%)	1123.0	0.40	0.093
Warm front	99 (6.7%)	64 (10.3%)	81 (12.1%)	182 (5.2%)	1994	0.23	0.12
Cold front	244 (16.5%)	159 (25.6%)	136 (20.4%)	233 (6.7%)	1792.5	0.75	0.033
All	1478	621	668	3483			

Fig. S16. Sensitivity of cold front intensity comparisons between urban and rural areas. Similar to Fig. 4a, the decreased intensity remains statistically significant with elevated reflectivity thresholds (25 and 45 dBZ).

Referee #4 (Remarks to the Author):

I co-reviewed this manuscript with one of the reviewers who provided the listed reports. We appreciate your inputs in helping us improve the manuscript quality.

Referee #3 (Remarks to the Author):

The authors have made the efforts to improve the quality of this study. I have some remaining comments.

We appreciate your constructive suggestions throughout the review process, which have helped us improve the quality of this study.

1. Line 511-517: The classification of storms in this study exhibits large uncertainties. I appreciate that the authors have checked the data in National Hurricane Center and found the discrepancy between “tropical cyclones” in previous version and the so-called “tropical systems” in this version. Only 14 hurricanes/tropical storms out of 26 events were “correctly” identified using the method. There might also be large uncertainties in cold or warm fronts. It seems that the authors did not expect such a significant discrepancy. That should motivate us to identify the storms of interest meteorologically and physically.

We appreciate this comment regarding uncertainties in storm classification. Using National Hurricane Center records and additional references, we find that among the 26 events identified as tropical systems, 14 were officially named hurricanes or tropical storms, and 8 were other weather systems influenced by tropical air flow (for example, tropical waves and tropical-influenced convective systems). We acknowledge that this discrepancy reflects uncertainties in our classification framework. However, our goal is not to strictly identify meteorological storm types, but to systematically group all events that produce heavy rainfall into five consistent categories for urban–rural comparison. This approach may introduce some mixing across more specialized storm types but allows sufficient sample sizes to identify urban effects. The current framework therefore represents a balance between physical interpretability and statistical robustness. To address this uncertainty, we have added a parenthetical caveat in the Sensitivity and Uncertainty Analyses section.

“Strictly speaking, this algorithm is based on storm characteristics than pre-defined storm type categories, although it aligns well with commonly recognized storm types. Rather than strictly identifying meteorological storm types, we group storm events into five basic categories for urban–rural comparison. This simplification may introduce some uncertainty and mixing across more specialized storm types. We expect this approach to be well suited for studying storm-urban interactions, as it relies on intrinsic storm properties that govern how storms respond to urban environments.”

2. The reflectivity should be specified when discussed, because the reflectivity data used here is in 3D. Did the authors use maximum values or average values of reflectivity at the vertical level? I couldn't find the description in the methodology of the manuscript. For example, in Tables S2 and S3, the authors defined the kinds of storms based on AR and AHR. What does 40 dBZ represent here, maximum and average reflectivity at that spatial grid? Clarification would be very useful here.

Thank you for this comment. We have revised this sentence to, *“the rainy area and heavy rain area are quantified as the largest two-dimensional contiguous regions with the column-maximum reflectivity exceeding 20 dBZ and 40 dBZ, respectively, throughout the storm passing through the observation window”* (L549–L551)

3. Lines 82-84: “selecting those with high reflectivity areas (40 dBZ..)”. It might be helpful to specify what reflectivity is discussed. At which level? Maximum value? Average value? This would avoid confusion when reading the other parts of the manuscript.

This sentence has been modified to, *“We focus on the storms that produce heavy rainfall over urban domains during the warm season (May–September), selecting events in which high*

reflectivity areas (*the column-maximum reflectivity exceeding 40 dBZ, approximately 10 mm/h rainfall intensity*) cover at least 20% of the urban surface.” (L82–L85)

4. Line 97: Please specify what reflectivity is used here. Same is at Line 126.

The reflectivity data used here includes all vertical level, as described in the manuscript, “*Despite their high frequency, single-cell storms, because of their short duration and limited spatial extent, contribute only 7.5% (Dallas) to 13.5% (Houston) of the total number of high reflectivity grid cells (≥ 40 dBZ) across all altitudes (Extended Data Table 3).*” (L96–L98)

Original L126: This has been modified to, “*Occurrence of high reflectivity grid cells (≥ 40 dBZ) contributed by five storm types, aggregated over 23 warm seasons across all altitudes (1995 – 2017).*” (L414–L415)

5. Lines 326-327: “We found more frequent local-scale single cell storms and isolated storms...”
Where did you find more frequent storms described here and in what context?

This sentence has been revised as, “*Local-scale single-cell storms and isolated storms occur more frequently over cities, particularly at night...*” (L273–L274)

6. Lines 266-267: It should be “The number of high reflectivity grid boxes increased as cold fronts approach the domains and decreased as they move away.”

The tense has been corrected as suggested.

7. Line 279: “before the city” might be “before the storm reaches the city”? Please clarify.

It has been changed as: “*The observed decrease in cold frontal storm intensity over urban areas, along with the potential enhancement ahead to the city...*” (L233–L234)

Referee #4 (Remarks to the Author):

I co-reviewed this manuscript with one of the reviewers who provided the listed reports.

We appreciate your inputs in helping us improve the manuscript quality.